# Rainfall thresholds estimation for shallow landslides in Peru from gridded daily data

Carlos Millán-Arancibia[1,2] and Waldo Lavado-Casimiro[1]

[1]National Service of Meteorology and Hydrology of Peru (SENAMHI), Lima, 15072, Peru
[2]Universidad Nacional Agraria La Molina (UNALM), Lima 15012, Peru

**Correspondence:** Carlos Millán (cmillan@senamhi.gob.pe)

**Abstract.** The objective of this work is to generate and evaluate regional rainfall thresholds obtain from a combination of high-resolution gridded rainfall data (PISCOpd_Op), developed by the National Service of Meteorology and Hydrology of Peru (SENAMHI), and information from observed shallow landslide events. The landslide data were associated with rainfall data, determining triggering and non-triggering rainfall events with rainfall properties from which rainfall thresholds are determined.

The validation of the performance of the thresholds is carried out with events that occurred during 2020 and focuses on evaluating the operability of these thresholds in landslide warning systems in Peru. The thresholds are determined for 11 rainfall regions. The method of determining the thresholds is based on an empirical–statistical approach, and the predictive performance of the thresholds is evaluated whit the "true skill statistics" (TSS). The best predictive performance is the mean daily intensity-duration ($I_{mean}-D$) threshold curve, follow by accumulated rainfall $E$. This work is the first attempt to estimate

regional thresholds on a country scale to better understand landslides in Peru, and the results obtained reveal the potential of using thresholds in the monitoring and forecasting of shallow landslides caused by intense rainfall and in supporting the actions of disaster risk management.

## 1 Introduction

Landslides are one of the most globally impactful hazards causing casualties and damage to public and private property, and

15 are responsible for at least 17% of all natural hazard deaths in the world (Chae et al., 2017; Segoni et al., 2018). Rain is the main trigger for shallow landslides, which are responsible for fatalities and economic losses worldwide (Petley, 2012). In Perú, landslides are the fifth most common natural hazard generating the most emergencies in the last 16 years (INDECI, 2019), along with heavy rains, low temperatures, strong winds, and floods. Most landslides occur during the South American monsoon (Zhou and Lau, 1998) between November and April, and most of them belong to the category of debris flow that is shallow

in nature (Naidu et al., 2018). However, consideration of the physiographic and climatic environment of the country with regard to the relationship between rainfall and landslides has not yet been investigated. Therefore, knowing and understanding the interrelationship between landslides and rainfall, its main trigger, could be valuable in objectively proposing warning and monitoring systems for areas susceptible to landslides. Terrain saturation is the original cause of landslide occurrence, and this saturation effect can arise in different ways (intense rains, thaws, changes in the level of groundwater, water discharge in lakes,

lagoons, and reservoirs, and an increase in flow in channels, streams, and rivers). Out of all these factors that cause saturation and affect soil stability conditions, rainfall is the most frequent and important one in triggering landslides (Prenner et al., 2018; Segoni et al., 2014). However, the maximum probability of occurrence of landslides is not always associated with extreme conditions of heavy rainfall and soil moisture; there is also the influence of the antecedent condition of rainy days prior to the occurrence of landslides (Abraham et al., 2020; Leonarduzzi et al., 2017).

One of the techniques used in the study of rainfall as a triggering factor for landslides is the determination of rainfall thresholds, which has been widely studied worldwide using various methods (empirical, statistical, manual, probabilistic methods, and with physically-based models) (Guzzetti et al., 2007; Segoni et al., 2018; Tang et al., 2019; Berti et al., 2020). For rain-induced landslides, the threshold can be defined as rainfall, soil moisture, or hydrological conditions that, when reached or exceeded, are likely to trigger landslides. Thresholds have been developed at different time (sub-hourly, hourly, daily, monthly)

and spatial scales (local, basin, regional, national, global) depending on the information available (Segoni et al., 2018). For example, there is been developed empirical–statistical approach to the estimation of global thresholds (Caine, 1980; Guzzetti et al., 2008; Kirschbaum and Stanley, 2018) and national thresholds (Leonarduzzi et al., 2017; Peruccacci et al., 2017; Uwihirwe et al., 2020). Empirical approaches to forecasting the occurrence of landslides depend on the definition of rainfall thresholds obtained from different hydrometeorological variables (Gariano et al., 2015; Segoni et al., 2018). There is a large number

of analysis variables that could be used to define thresholds (up to 22 variables were reported) (Guzzetti et al., 2007, 2008). Under this approach, rainfall thresholds aim to separate the rainfall events that triggered landslides from those rainfall events that did not result in landslides. This empirical approach is widely applied because its analysis and implementation do not require the constant monitoring of the other physical variables on which other types of most robust models are based (e.g., physically-based models), and this drawback of the robust models is the main advantage of empirical approaches and its ap-

plicability over large areas (Rosi et al., 2012). Another advantage for its application is that it is not subject to the challenges accompanied with other models, mainly the many high-quality input data, such as soil information that is needed, which is associated with high uncertainties too.

     Thresholds can be set for different spatial scales depending on the extent of the analysis, and these can be categorized into six classes: global, national, regional, basin, local, and slope scales. A regional scale is understood to be the administrative

subdivision of a nation, typically extending over thousands of square kilometers (Segoni et al., 2018). In the study of national territories, it is necessary to take into account the high meteorological and spatial physiographic variability of the study area, in order to obtain more accurate and reliable rainfall thresholds. This is achieved through the regionalization of the study area into areas with homogeneous meteorological conditions (Segoni et al., 2014). Regionalization in the analysis of thresholds associated with landslides has been used with different approaches; for example, rainfall indices have been used, such as

the annual average, daily maximum, monthly average, and monthly daily maximum rainfall, among others (Augusto Filho et al., 2020; Segoni et al., 2014), as well as an environmental subdivision within a national territory based on erodibility and climatology represented by the maximum daily intensity of a rainfall event (Leonarduzzi et al., 2017) or on topography, lithology, land-use, land cover, climate, and meteorology (Peruccacci et al., 2017). In this study, we refer to regions, such as the subdivision of the Peruvian territory, from a maximum daily rainfall perspective.

The main objective of this work is to estimate rainfall thresholds for the monitoring of shallow landslides generated by rainfall from a gridded rainfall database. Additionally, this work focuses on implementing an objective methodology for landslide monitoring that is based on observed landslide events. The novelty of this work is that this is the first approximation of rainfall thresholds in Peru that combines gridded rainfall data and observed event data for landslide monitoring.

## 2    Materials and methods

### 2.1    Area of study

Peru is located on the west coast of South America and is characterized by maximum rainfall rates that occur between November and March in its Andean region, with most of the rainfall being produced by convection (Lavado Casimiro et al., 2011). Peru's climate variability is determined by the South American monsoon system, the southward shift of the Intertropical Convergence Zone (ITCZ), and differential warming between the ocean and the land, which contributes to a greater influx
of moisture eastward from the tropical Atlantic Ocean to the South American continent, and in which the Andes mountain range plays an important role modulating rainfall on both the eastern and western slopes (Poveda et al., 2014; Bookhagen and Strecker, 2008; Boers et al., 2014; Lavado Casimiro et al., 2011; Llauca et al., 2021).

This study adopts the study domain defined for the Monitoring System of Potential Mass Movements Generated by Heavy Rains (SILVIA) (Millan, 2020; Millan et al., 2021) of the National Service of Meteorology and Hydrology of Peru (SENAMHI).
This domain was obtained from the superposition of two databases. The first one was a map of landslide susceptibility from the Geological, Mining and Metallurgical Institute of Peru (Villacorta et al., 2012), which has five categories of susceptibility. The second database contained information regarding spatial discretization in basins of the GEOGloWS ECMWF Streamflow Service (David et al., 2011; Qiao et al., 2019; Souffront Alcantara et al., 2019; Lozano et al., 2021), from which the domain of this study was discretized in 5373 basins with median areas of approximately 105 km2. The study area and spatial distribution
of the basins are shown in Figure 1.

### 2.2    Rainfall data: PISCOpd_Op

The main source of information for this study was the gridded daily rainfall dataset PISCOpd_Op (Gridded Daily Rainfall Operative data of PISCO). PISCOpd_Op is an operational rainfall dataset part of the Peruvian Interpolated data of SENAMHI's Climatological and Hydrological Observations (PISCO) with gridded data on rainfall (Aybar et al., 2020), air temperature
(Huerta et al., 2018), reference evapotranspiration (Huerta et al., 2022) and monthly discharges (Llauca et al., 2021) at the scale of all of Peru. PISCOpd_Op has a spatial resolution of 0.1° and a daily temporal resolution. PISCOpd_Op has data from 1981 and is updated daily, accumulating daily rainfall (from 7 a.m. to 7 a.m.), generated from   416 conventional SENAMHI rain gauge network (see Figure 1). PISCOpd_Op is generated based on a genRE interpolation method (van Osnabrugge et al., 2017), which consists of an interpolation using inverse distance weighting (IDW) and includes multipliers that are based on
the monthly climatology of PISCOp.

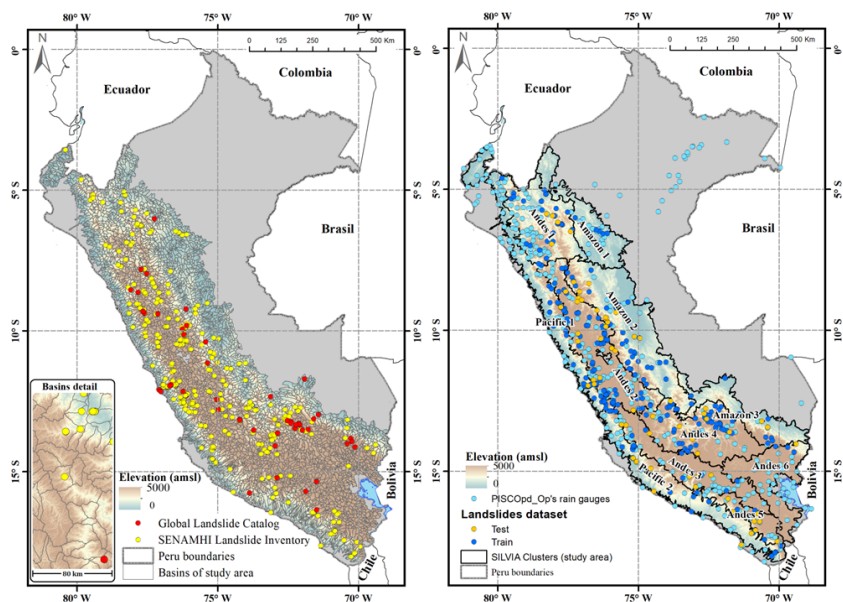

**Figure 1.** Study area. Left: Spatial distribution of the Global Landslide Catalog (red) and SENAMHI landslide inventory (yellow). Right: Eleven landslide-susceptibility regions for Peru and distribution of calibration (blue) and validation (yellow) landslides.

## 2.3 Landslide event data

The second main source of information used for this research was two inventories of observed and collected landslide events: SENAMHI's of Rainfall-Triggered Shallow Landslides Inventory of Peru (SLIP) and NASA's Global Landslide Catalog (GLC) (Kirschbaum et al., 2015a). Both catalogs consider all types of shallow landslides triggered by rainfall that have been reported in the media, in databases of agencies associated with disasters, in scientific reports, and other available sources. Most of them belong to the debris flow category which is shallow in nature (Naidu et al., 2018). In this sense, this study used shallow landslide (SL) for all types of shallow landslide processes.

SLIP was implemented in January 2019 and has 330 records from the 2014–2020 period. Therefore, there is a greater degree of certainty regarding the number of events recorded in recent years. It should be noted that this inventory was implemented in January 2019. Therefore, there is a greater degree of certainty regarding the number of events recorded in recent years. The GLC has 6788 registrations for the whole world; while for Peru, 49 landslide events have been registered, which were temporally distributed between 2007 and 2014. For the use of these data, exploratory analyses were performed to avoid inconsistencies in the recording of the events. The spatial correspondence of the data was evaluated through geospatial analysis of points and areas in the study area, and the registration information was subsequently excluded or corrected. We also assessed data consistency with regard to typographical errors. As a result, two incongruous events were determined: The first one was reported in a place without landslide occurrence conditions and was therefore not considered in the analysis. In the second

event, an error in its spatial tabulation was determined; this error was corrected, and the event was included in the analysis. The total number of landslide records is 377, and the spatial distribution of these events is shown in Figure 1.

## 2.4   Rainfall threshold model

An empirical–statistical approach was used to define rainfall thresholds for landslide-susceptible regions, consisting of the following steps: (1) determination of rainfall events from a historical rainfall series, (2) definition of the variables of rainfall events, (3) define landslides regions from maximum daily rainfall region and GEOGloWS basins for the area studio, (4) threshold estimation for individual rainfall event variables for calibration period based on an objective maximization of predictive performance, (5) threshold estimation for combination of rainfall event variables for calibration period based on an 115   objective maximization of predictive performance, and (6) run thresholds models and get metrics for analysis and discussions (methodology is presented in Figure 2). Below are the details of the method.

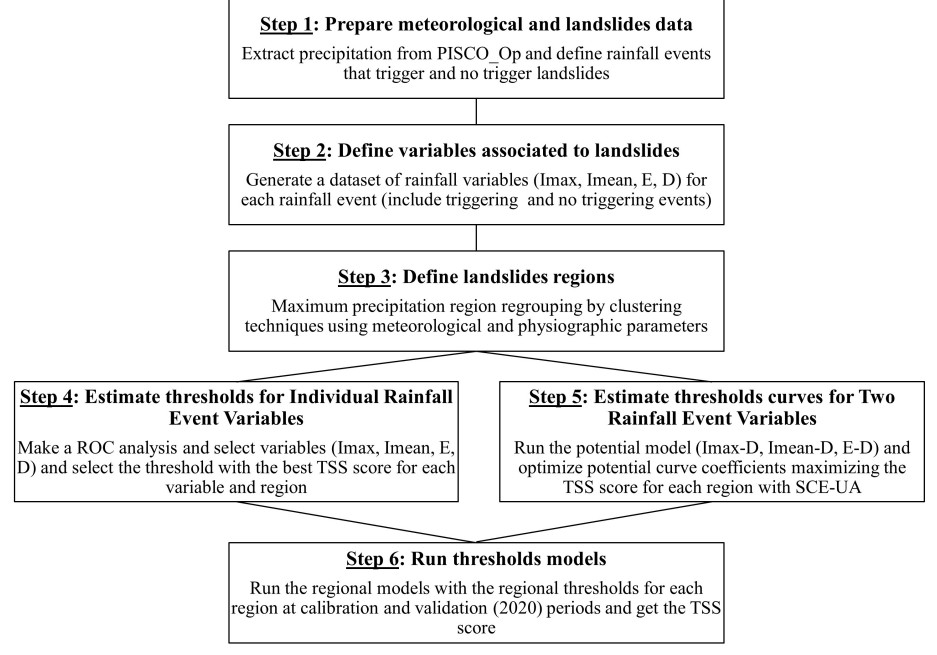

**Figure 2.** Methodology six steps.

The first step was the construction of a historical rainfall series from gridded rainfall data (PISCOpd_Op) for each basin that had a minimum of one landslide event. After obtaining the rainfall series, rainfall events were defined along with a historical series for each selected basin. For this work, we define an independent rainfall event as a series of consecutive rainy days 120   where it has rained above a minimum rainfall threshold (Figure 3). Many authors use minimum thresholds of 1 mm to define rainy days (Dai, 2006; Dai et al., 2007; Han et al., 2016; Leonarduzzi et al., 2017; Shen et al., 2021; Tian et al., 2007; Yong et al., 2010). However, given the great climatological spatial variability in the study area, it was determined that there was not

a single minimum threshold for the entire territory, but a minimum threshold was discretized from the bias of PISCOpd_Op for non-rainy days. The PISCOpd_Op bias was determined when rain gauges did not report rain (0 mm), and the discretized minimum threshold (Umin) of rain was defined according to the following Equation 1:

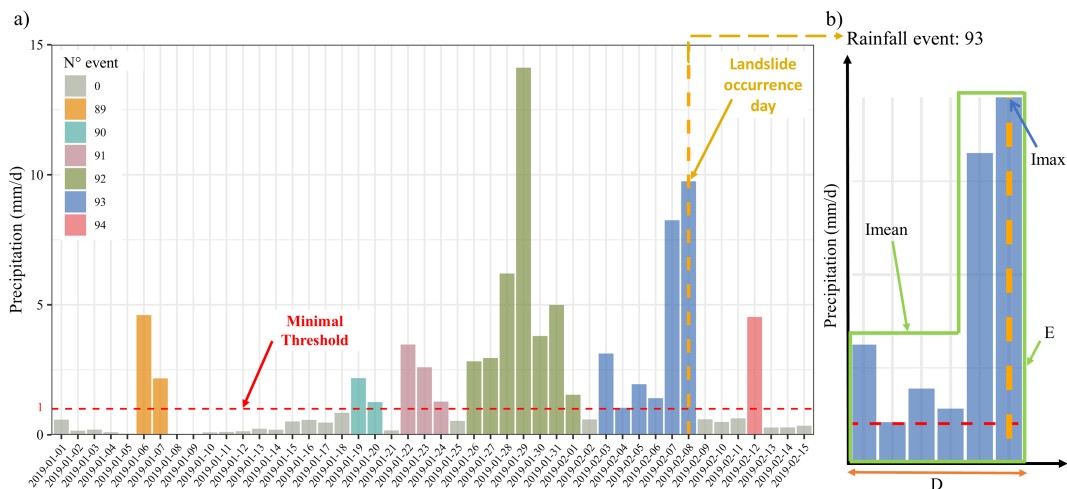

**Figure 3.** a) Extract from the rainfall time series (rainy period 2019) for an example basin, where the estimated rainfall events are observed (each color is a rainfall event, the lead-colored event 0 is the non-rainy days). b) An example of a rain event associated with the occurrence of a landslide, in this case the rain event No. 93, where the variables analyzed for the estimation of thresholds are shown: the maximum daily intensity $I_{max}$ (mm/day), the accumulated rainfall $E$ (mm), the duration $D$ (day), and the mean daily intensity $I_{mean} = E/D$ (mm/day).

$$U_{min} = \begin{cases} U_0 & \text{if } s \leq U_0 \\ s & \text{if } s > U_0 \end{cases} \qquad (1)$$

where $s$ is the average of simple bias when rainfall stations reported a value of 0 rainfall compared with the estimation in PISCOpd_Op. And $U_0$ is the initial minimum rainfall threshold, and it is established as 1 mm for all regions with exception of coastal Pacific regions which is considered 0.5 mm. Once rainfall events were defined, whether they were triggering or

non-triggering events were established. A rainfall event is considered a rainfall trigger event if it is associated with a landslide event, i.e. if throughout the rainfall event duration a shallow landslide has occurred.

The second step was to determine analysis variables for each rainfall event, for which the maximum daily intensity $I_{max}$ (mm/day), the accumulated rainfall $E$ (mm), the duration $D$ (day), and the mean daily intensity $I_{mean} = E/D$ (mm/day) were calculated. Concerning the triggering rain events, two scenarios were considered. For the first scenario (entire event - EE), the

properties of the rainfall event (Figure 3) were defined considering the rainfall rate of the landslide occurrence day. The second scenario (antecedent event - AE) defined the properties up to one day before the occurrence, i.e., it did not consider the rainfall rate of the landslide occurrence day. The reason for analyzing the second scenario was to evaluate the level of incidence that

is attributed only to antecedent conditions for landslide occurrence, as this allows us to evaluate if it is possible to forecast or warn landslides based only on the antecedent conditions. The temporal evolution of hydrometeorological variables provides an idea of how the critical conditions of the activation of landslides develop (Prenner et al., 2018; Segoni et al., 2018).

The third step consisted in divide the study area into regions based on clustering techniques (this step is explained in more detail in section 2.5). Next, GEOGloWS basins were merged with regions to determine their spatial correspondence. The fourth and fifth step was to objectively select a rainfall threshold that separates triggering rainfall events from non-triggering rainfall events with the best level of predictive performance. Rainfall thresholds were established by maximizing predictive performance in two ways: the first way includes every rainfall event property independently ($I_{max}, E, D, I_{mean}$), and the second one determined was through curve-like thresholds that related two properties ($I_{max} - D, E - D, I_{mean} - D$) in the form of $V = a.D^{-b}$, where $V$ represents the rainfall properties ($I_{max}, E, D, I_{mean}$); a and b are the scale and shape parameters of the curve (while for logarithmic space, a is the intersection parameter and b denotes the slope of the linear curve). The approximation of the thresholds based on only one of the rainfall event properties ($I_{max}, E, D$ or $I_{mean}$), was estimated whit the minimum radial distance to the perfect classificatory test ($TSS = 1$, with $se = 1$ and $1 - sp = 0$) from the ROC space (e.g., Uwihirwe et al., 2020; Postance et al., 2018; Gariano et al., 2015) and the approximation of curve-like thresholds, was established with the maximum true skill statistic based on the optimization of $a$ and $b$ parameters of the curve model $V = a.D^{-b}$ with an initial approximation of the curve based on $a$=average of the variable V of the triggering rainfall events and $b = 0.5$. Finally, the sixth step consisted in apply the model to the rainfall events and compare with the observed landslides events and get the predictive performance metrics for each region at calibration and validation periods.

## 2.5 Regionalization

According to the study, on a national scale, it is necessary to consider the meteorological and spatial physiographic high variability governing the country to obtain reliable rainfall thresholds, since a single global or national threshold cannot represent such variability. To achieve rainfall thresholds on a national scale, the approach used was the regionalization of the study area in areas with homogeneous meteorological conditions (Segoni et al., 2014). Research related to thresholds have used rainfall indices such as the annual average, daily maximum, monthly average, monthly daily maximum of rainfall, and other environmental variables for the regionalization of study areas (Augusto Filho et al., 2020; Leonarduzzi et al., 2017; Segoni et al., 2014).

This study uses SENAMHI's Homogeneous Regions of Maximum Daily Rainfall (Yupanqui et al., 2017) as input for the regionalization of the study area. These regions were determined based on clustering techniques from rainfall information from 535 automatic stations, in which 10 macroregions and 30 subregions of maximum daily rainfall were natively identified. The climatic regions established for the present study consisted of a grouping of the 30 maximum daily rainfall regions. The regrouping consisted of a multi-criteria analysis based mainly on the fact that the grouped regions did not exceed a threshold value of 10 in the heterogeneity test (Hosking and Wallis, 1997), which included events recorded in the databases in addition to sharing the similarity of the covariates of relief (altitude) and climatology (mean rainfall). Although this value of 10 indeed exceeds the level of heterogeneity recommended in 2, this tolerance is contemplated since they are regions obtained from a

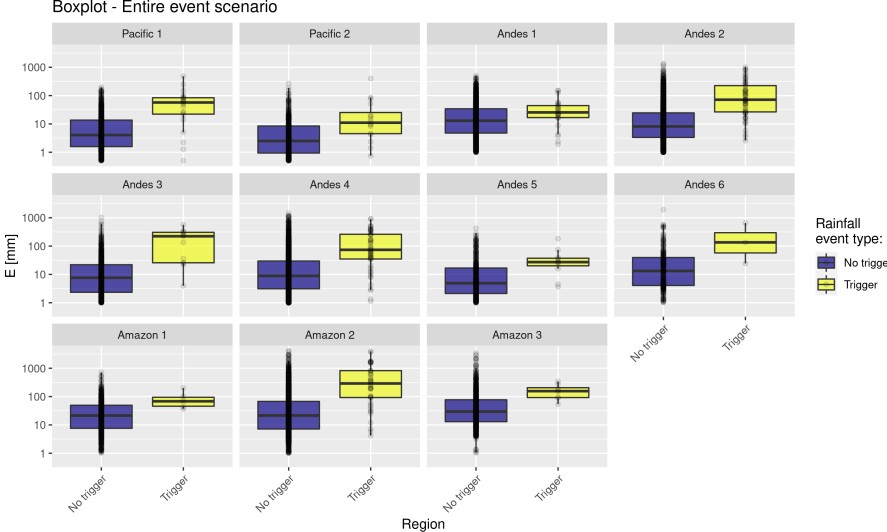

**Figure 4.** Boxplot of triggering (yellow) and no triggering (blue) total cumulative rainfall E for the eleven regions established in this study for Peru. The boxplot graphs include outliers and show the potential predictive for the E variable to separate the rainfall events that trigger/no trigger shallow landslides. Also, the plot shows the regional variability of the rainfall events that trigger shallow landslides.

regrouping. From this analysis, 11 regions were obtained for the study area (see Figure 1). Four thresholds of independent variables ($I_{max}, E, D, I_{mean}$) and three curved thresholds ($I_{max} - D, E - D, I_{mean} - D$) were defined for each region. The total was 77 thresholds for the study area, and 7 thresholds for each region. Figure 4 presents an accumulated rain $E$ box
diagram showing its predictive power to discriminate between triggering and non-triggering rainfall events.

## 2.6   Calibration and validation of thresholds

Calibration and validation are fundamental processes for objectively defining thresholds. The purpose of calibration is to estimate thresholds based on the maximization of predictive or classifier performance capacity. Validation aims to show the potential of the ability to predict or differentiate those rainfall events that trigger landslides. Among the calibration and vali-
dation approaches, the most recommended is to divide the datasets for threshold estimation and another independent set for validation (Segoni et al., 2018). In this work, 377 recorded landslide events were used to define rainfall thresholds in Peru (Figure 1). For the calibration, all events occurring before 2020 were selected, representing approximately 70% of the recorded events. Regarding the validation process, it consisted of evaluating thresholds calibrated using the landslide events recorded in 2020, which represented approximately 30% of the recorded events. This process was carried out for the year 2020, as we
wanted to know how the thresholds would perform when they were assimilated into a regional early warning system. This method of calibration/validation that set one year of the dataset to validation is a method that has been used in other research (e.g., Kirschbaum et al., 2015b; Dikshit et al., 2019).

For the evaluation of the thresholds in calibration and validation was used a confusion matrix (also called a contingency table). The confusion matrix is a tool used to determine the accuracy of binary classification models (triggering and non-triggering rainfall events), and also, used to evaluate the analysis of concordance between the results of the model and the observed data. A confusion matrix was computed for each threshold and counted the number of true successes or true positives (TP), the number of false positives (FP), the number of true negatives (TN), and the number of false negatives (FN) (Figure 5). From which various performance statistics can be calculated. Some of the most common measures for landslide forecasting are the sensitivity ($s_e = TP/(TP+FN)$), specificity ($s_p = 1-FP/(FP+TN)$) and true skill statistic ($TSS = s_e + s_p - 1$) (e.g., Staley et al., 2013; Gariano et al., 2015; Leonarduzzi et al., 2017; Mirus et al., 2018; Leonarduzzi and Molnar, 2020; Hirschberg et al., 2021).

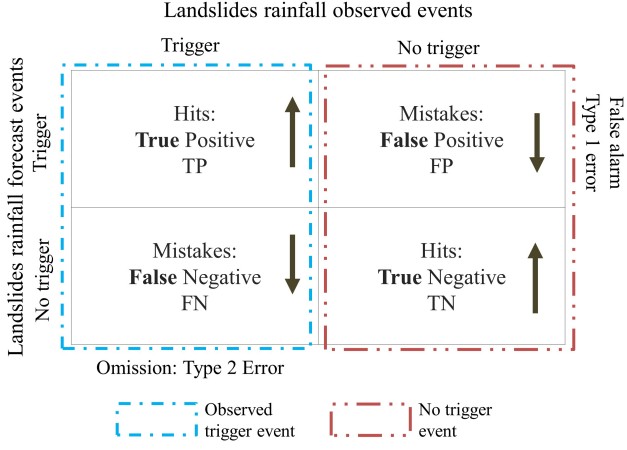

**Figure 5.** Confusion matrix definition for classification model.

The TSS is an efficiency statistic that helps in the measurement of the goodness-of-threshold models, as it is an integrative measure of the predictive performance of the model. The TSS is more objective than simply a random manual estimate (Frattini et al., 2010). It varies between 1 and –1, with its optimal score equal to 1, which indicates the maximum performance of the model. TSS=$s_e$-(1-$s_p$) is the difference between the true positive rate (sensitivity $s_e$) and false alarm rate (1-specificity $s_p$), which are the two most important components for providing early warnings (Leonarduzzi et al., 2017). The TSS is also referred as the Peirce skill score (Peirce, 1884), the Youden index (Youden, 1950), or the Hanssen–Kuipers skill score (Hanssen and Kuipers, 1965). The benefit of using the specificity over the false positive rate (FPR=FP/(FP+TN)) is that in a perfect model TSS, sensitivity and specificity all equal 1 (Hirschberg et al., 2021).

For thresholds based on rainfall event properties independently ($I_{max}$, $E$, $D$ or $I_{mean}$), the overall impression of the predictive power was estimated whit the so-called receiver operating characteristic (ROC) curve (Fawcett, 2006), from which the minimum radial distance to the perfect classificatory test (TSS=1, with $s_e$=1 and 1-$s_p$=0) was used to select the individual variable threshold (e.g., Uwihirwe et al.; Gariano et al.; Postance et al.) while for the threshold curve ($I_{max}-D$, $E-D$, $I_{mean}-D$) the scale parameter a and the shape parameter b are simultaneously tuned to maximize the true skill statistics (TSS) (e.g.,

**Table 1.** Rainfall thresholds of independent variables (Th: threshold, Rad:Minimum radial distance, Cal: Calibration, Val: Validation)

| Scenario | Region | E | | | | I_mean | | | | Imax | | | | D | | | |
|---|---|---|---|---|---|---|---|---|---|---|---|---|---|---|---|---|---|
| | | | | TSS | | | | TSS | | | | TSS | | | | TSS | |
| | | Th | Rad | Cal | Val | Th | Rad | Cal | Val | Th | Rad | Cal | Val | Th | Rad | cal | Val |
| Entire event | Pacific 1 | 21.16 | 0.38 | 0.66 | 0.20 | 5.62 | 0.33 | 0.54 | 0.60 | 10.11 | 0.43 | 0.68 | 0.58 | 8 | 0.49 | 0.56 | -0.09 |
| | Pacific 2 | 4.23 | 0.48 | 0.44 | 0.39 | 2.12 | 0.35 | 0.61 | 0.20 | 4.55 | 0.34 | 0.51 | 0.27 | 7 | 0.48 | 0.30 | 0.39 |
| | Andes 1 | 16.15 | 0.49 | 0.39 | 0.12 | 6.20 | 0.41 | 0.43 | 0.18 | 11.84 | 0.43 | 0.38 | 0.23 | 2 | 0.60 | 0.19 | -0.16 |
| | Andes 2 | 23.92 | 0.29 | 0.58 | 0.41 | 5.17 | 0.30 | 0.51 | 0.28 | 8.59 | 0.29 | 0.58 | 0.47 | 8 | 0.31 | 0.54 | 0.33 |
| | Andes 3 | 25.35 | 0.21 | 0.78 | 0.41 | 6.01 | 0.19 | 0.83 | 0.22 | 16.72 | 0.08 | 0.92 | 0.34 | 21 | 0.29 | 0.69 | 0.28 |
| | Andes 4 | 38.85 | 0.35 | 0.51 | 0.61 | 6.17 | 0.42 | 0.45 | 0.61 | 8.44 | 0.40 | 0.43 | 0.69 | 9 | 0.43 | 0.45 | 0.33 |
| | Andes 5 | 25.52 | 0.27 | 0.67 | 0.39 | 4.25 | 0.35 | 0.52 | 0.37 | 9.75 | 0.29 | 0.61 | 0.51 | 4 | 0.33 | 0.54 | 0.03 |
| | Andes 6 | 24.32 | 0.36 | 0.64 | 0.66 | 4.05 | 0.57 | 0.40 | 0.44 | 5.56 | 0.54 | 0.45 | 0.46 | 6 | 0.31 | 0.68 | 0.69 |
| | Amazon 1 | 37.20 | 0.36 | 0.64 | - | 12.68 | 0.26 | 0.74 | - | 20.73 | 0.34 | 0.66 | - | 3 | 0.57 | 0.29 | - |
| | Amazon 2 | 92.77 | 0.31 | 0.57 | 0.52 | 8.88 | 0.44 | 0.41 | 0.34 | 16.15 | 0.41 | 0.46 | 0.38 | 5 | 0.40 | 0.51 | 0.37 |
| | Amazon 3 | 53.99 | 0.32 | 0.68 | 0.66 | 11.14 | 0.59 | 0.41 | 0.39 | 17.74 | 0.48 | 0.52 | 0.55 | 12 | 0.50 | 0.44 | -0.10 |
| Antecedent event | Pacific 1 | 19.01 | 0.18 | 0.63 | 0.35 | 4.87 | 0.30 | 0.51 | 0.90 | 10.11 | 0.22 | 0.65 | 0.91 | 7 | 0.28 | 0.60 | -0.11 |
| | Pacific 2 | 18.60 | 0.27 | 0.53 | -0.17 | 2.98 | 0.33 | 0.43 | -0.21 | 10.56 | 0.35 | 0.43 | -0.11 | 6 | 0.28 | 0.49 | 0.55 |
| | Andes 1 | 7.57 | 0.65 | 0.14 | 0.42 | 5.70 | 0.47 | 0.35 | 0.63 | 7.57 | 0.54 | 0.30 | 0.57 | 7 | 0.73 | 0.04 | -0.11 |
| | Andes 2 | 40.03 | 0.28 | 0.59 | 0.42 | 5.26 | 0.29 | 0.51 | 0.30 | 9.74 | 0.27 | 0.59 | 0.33 | 7 | 0.32 | 0.54 | 0.38 |
| | Andes 3 | 127.47 | 0.30 | 0.69 | 0.53 | 6.08 | 0.23 | 0.55 | 0.33 | 16.72 | 0.16 | 0.77 | 0.44 | 20 | 0.29 | 0.69 | 0.34 |
| | Andes 4 | 31.73 | 0.33 | 0.53 | 0.57 | 5.77 | 0.37 | 0.51 | 0.59 | 8.44 | 0.35 | 0.50 | 0.60 | 9 | 0.40 | 0.46 | 0.24 |
| | Andes 5 | 15.77 | 0.31 | 0.57 | 0.32 | 2.22 | 0.56 | 0.43 | 0.28 | 8.25 | 0.31 | 0.57 | 0.31 | 3 | 0.41 | 0.44 | 0.01 |
| | Andes 6 | 18.76 | 0.43 | 0.55 | 0.60 | 3.75 | 0.60 | -0.14 | 0.41 | 4.75 | 0.64 | 0.33 | 0.44 | 5 | 0.38 | 0.61 | 0.67 |
| | Amazon 1 | 70.79 | 0.54 | 0.31 | - | 10.18 | 0.48 | 0.32 | - | 13.26 | 0.52 | 0.34 | - | 15 | 0.83 | 0.14 | - |
| | Amazon 2 | 175.88 | 0.36 | 0.53 | 0.64 | 8.81 | 0.44 | 0.44 | 0.40 | 16.15 | 0.40 | 0.48 | 0.45 | 17 | 0.39 | 0.51 | 0.47 |
| | Amazon 3 | 137.64 | 0.52 | 0.35 | 0.30 | 11.05 | 0.59 | 0.41 | 0.39 | 16.90 | 0.49 | 0.51 | 0.53 | 11 | 0.50 | 0.10 | -0.10 |

Leonarduzzi et al.; Hirschberg et al.). This maximization was automatically calibrated using the shuffled complex evolutionary algorithm (SCEA-UA) (Duan et al., 1993), considering the TSS as the objective function. The methodology was applied for each region within the analysis area, finding different thresholds for each of them.

## 3 Results

### 3.1 Rainfall–landslide threshold

The calibrated thresholds for the individual properties of the events ($I_{max}, E, D, I_{mean}$) are shown in Table 1 and the curved thresholds ($I_{max} - D, E - D, I_{mean} - D$) are shown in Table 2. They are presented for two scenarios: the first one describes the rainfall events that include rainfall on the landslide occurrence day, called the entire event (EE); and the second one only includes the antecedent conditions up to one day before the landslide occurrence, called the antecedent event (AE), given that we are interested in analyzing landslide events under an approach that includes the predictive capacity of antecedent conditions

and their influence on the occurrence of future events for the operation of early warning services.

From the results, it is observed that thresholds with the best average performance for entire events were $E$ (TSS = 0.59) for individual properties and $I_{mean} - D$ (TSS = 0.65) for combined curves. As expected, the integration of properties into curves produced a better overall performance compared with the properties of individual events. Of the three curves ($I_{max} - D, E - D, I_{mean} - D$), the $I_{mean} - D$ curve performed the best (Figure 6), with TSS = 0.65 for calibration and TSS = 0.42

for validation.

The results show that the components with the lowest performance for threshold determination were duration ($D$) for both the calibration period and validation, followed by the average rainfall rate ($I_{mean}$). In the case of the combined curves, there is a smaller difference in their performances, with the $E - D$ being the one with the lowest performance. These thresholds do not have a good ability to discriminate landslide-triggering rainfall events of non-triggers.

### 3.2 Impact of regionalization

The study area was regionalized into 11 regions based on maximum daily rainfall information. The estimated results show the rainfall variability of Peru in the magnitudes of the thresholds for each region is presented in Table 1. Regionally, the best performing threshold of a single variable, cumulative rainfall E, averaging 33 mm, ranged from 4.23 mm (Pacific 2 region) to 92.77 mm (Amazon 2 region). $I_{max}$ ranged from 4.55 mm/d (Pacific 2 region) to 20.73 mm/d (Amazon 1 region) with

235 an average of 11.83 mm/d. The region with the best predictive performance was Andes 3 with a TSS of 0.8 for the mean of the thresholds of individual variables, and TSS of 0.89 for the mean of the threshold-type curve in scenario 2. The threshold with the best performance for this region was $I_{max}$ = 16.72 mm/d (TSS = 0.92), which correctly separated 100% of rainfall-triggering events and only had an 8% rate of false alarms. Similarly, the $I_{max} - D$ curve (TSS = 0.91) correctly separated 100% of rainfall-triggering events and only had a 9% rate of false alarms. A summary of the best single variable or curved thresholds

for each region is presented in Table 3.

Regionalization achieves a better separation of trigger and non-trigger distributions. The results for single-variable thresholds are presented in Figure 7. The calibrated thresholds performed better overall in the Andes 3 (TSS = 0.83) areas compared with the Andes 1 (TSS = 0.4), Andes 4 (TSS = 0.47), and Amazon 1 (TSS = 0.5) regions, which were the regions with the lowest performance. In fact, most of the landslides recorded occurred in the Andes 3 region (Figure 8). With respect to the two Pacific

regions, the Pacific 1 region (TSS = 0.66) performed better than the Pacific 2 region (TSS = 0.51). In the wettest regions of

**Table 2.** Rainfall thresholds of two variables (Th: threshold, Cal: Calibration, Val: Validation)

| Scenario | Region | $I_{mean}$-D | | | | $I_{max}$-D | | | | E-D | | | |
|---|---|---|---|---|---|---|---|---|---|---|---|---|---|
| | | Thresh | | TSS | | Thresh | | TSS | | Thresh | | TSS | |
| | | a | b | Cal | Val | a | b | Cal | Val | a | b | Cal | Val |
| | Pacific 1 | 11.55 | -0.44 | 0.68 | 0.26 | 16.73 | -0.17 | 0.71 | 0.28 | 27.92 | -0.16 | 0.66 | 0.21 |
| | Pacific 2 | 2.10 | -0.00 | 0.61 | 0.20 | 4.58 | -0.00 | 0.51 | 0.27 | 4.54 | -0.10 | 0.44 | 0.38 |
| | Andes 1 | 7.34 | -0.10 | 0.44 | 0.19 | 20.97 | -0.98 | 0.36 | 0.09 | 18.30 | -0.13 | 0.39 | 0.11 |
| | Andes 2 | 14.28 | -0.53 | 0.62 | 0.28 | 13.62 | -0.17 | 0.64 | 0.34 | 150.75 | -0.59 | 0.57 | 0.34 |
| | Andes 3 | 10.84 | -0.25 | 0.89 | 0.33 | 16.77 | -0.01 | 0.91 | 0.34 | 27.47 | -0.12 | 0.77 | 0.57 |
| Entire event | Andes 4 | 25.69 | -0.81 | 0.52 | 0.68 | 44.51 | -0.66 | 0.49 | 0.70 | 132.84 | -0.56 | 0.48 | 0.61 |
| | Andes 5 | 16.68 | -0.77 | 0.66 | 0.39 | 15.08 | -0.25 | 0.64 | 0.38 | 45.36 | -0.41 | 0.66 | 0.26 |
| | Andes 6 | 16.93 | -0.81 | 0.62 | 0.63 | 19.25 | -0.69 | 0.56 | 0.63 | 117.52 | -0.90 | 0.65 | 0.67 |
| | Amazon 1 | 14.25 | -0.05 | 0.77 | - | 20.91 | -0.02 | 0.66 | - | 37.89 | -0.03 | 0.64 | - |
| | Amazon 2 | 42.06 | -0.54 | 0.57 | 0.53 | 66.35 | -0.56 | 0.57 | 0.48 | 206.71 | -0.73 | 0.58 | 0.44 |
| | Amazon 3 | 36.74 | -0.45 | 0.73 | 0.68 | 49.54 | -0.42 | 0.73 | 0.70 | 54.10 | 0.00 | 0.68 | 0.66 |
| | Pacific 1 | 8.50 | -0.50 | 0.68 | 0.84 | 18.60 | -0.28 | 0.67 | 0.44 | 156.39 | -0.67 | 0.67 | -0.06 |
| | Pacific 2 | 14.85 | -0.88 | 0.53 | -0.17 | 25.15 | -0.31 | 0.47 | -0.08 | 34.46 | -0.37 | 0.53 | 0.19 |
| | Andes 1 | 6.45 | -0.08 | 0.36 | 0.66 | 7.52 | 0.00 | 0.30 | 0.56 | 9.21 | -0.03 | 0.18 | 0.46 |
| | Andes 2 | 11.54 | -0.39 | 0.65 | 0.39 | 19.37 | -0.54 | 0.60 | 0.43 | 113.10 | -0.53 | 0.58 | 0.31 |
| | Andes 3 | 13.98 | -0.26 | 0.80 | 0.48 | 16.01 | -0.49 | 0.73 | 0.48 | 387.59 | -0.37 | 0.69 | 0.54 |
| Antecedent event | Andes 4 | 19.29 | -0.72 | 0.56 | 0.66 | 34.81 | -0.69 | 0.51 | 0.66 | 31.59 | -0.00 | 0.53 | 0.57 |
| | Andes 5 | 8.59 | -0.63 | 0.53 | 0.41 | 23.61 | -0.66 | 0.62 | 0.22 | 45.51 | -0.67 | 0.60 | 0.06 |
| | Andes 6 | 16.39 | -0.92 | 0.55 | 0.59 | 18.54 | -0.89 | 0.51 | 0.57 | 83.96 | -0.93 | 0.61 | 0.64 |
| | Amazon 1 | 51.63 | -0.56 | 0.43 | - | 49.46 | -0.17 | 0.37 | - | 69.73 | 0.00 | 0.30 | - |
| | Amazon 2 | 22.41 | -0.42 | 0.53 | 0.51 | 33.70 | -0.32 | 0.54 | 0.49 | 388.25 | -0.30 | 0.53 | 0.64 |
| | Amazon 3 | 16.81 | -0.14 | 0.55 | 0.55 | 16.83 | -0.01 | 0.50 | 0.53 | 485.20 | -0.53 | 0.39 | 0.35 |

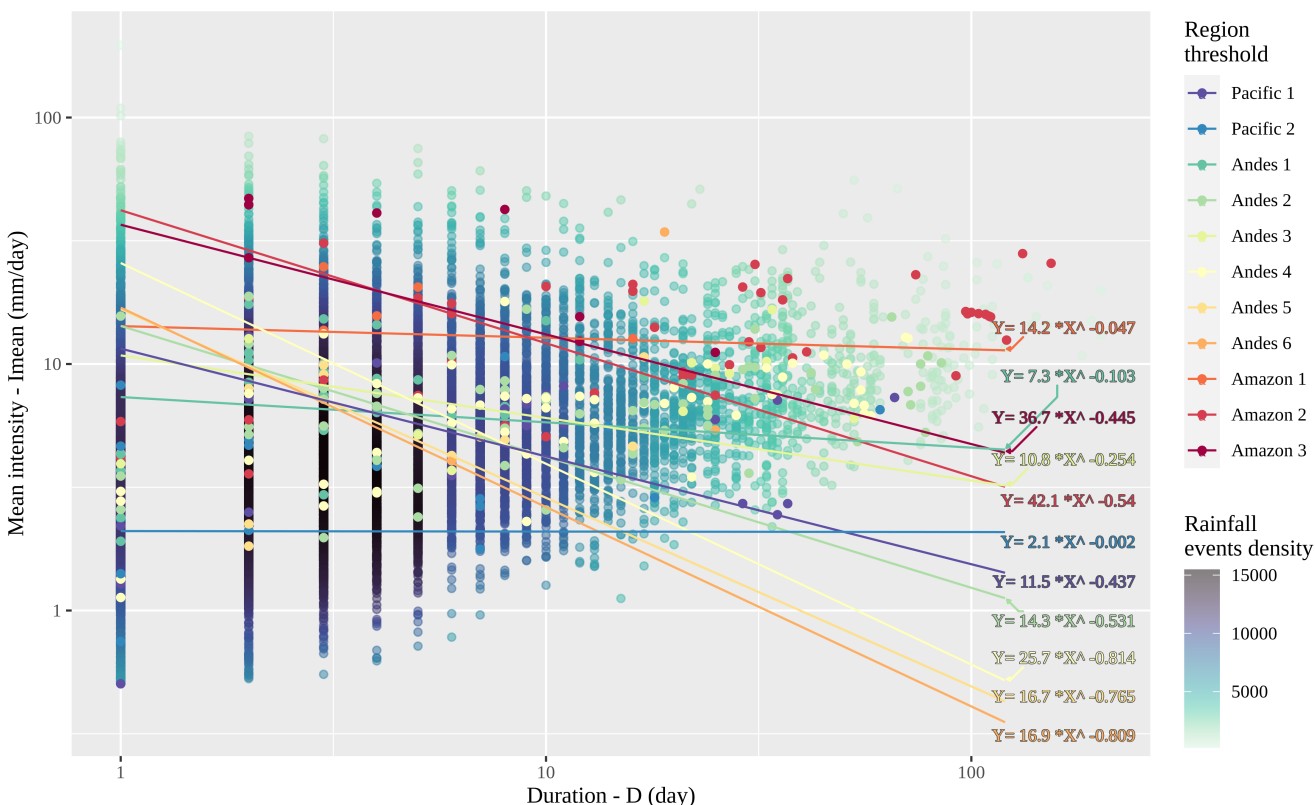

**Figure 6.** Mean Intensity-duration ($I_{mean} - D$) plots with regional threshold curves at logarithmic scale. The background with colored dots on a green-blue-black scale shows the density of rainfall events that do not trigger landslides. The rainfall events that trigger landslides were plotted with the same regional threshold color.

the Amazon, the Amazon 1 region was the best performing, followed by the Amazon 3 and Amazon 2 regions. This Amazon region and the Altiplano region (Andes 6) were the regions with the least calibration events.

The results do not show that any drainage (Pacific, Andes, or Amazon) stands out in separating triggering rain events from those that are not triggering; on the contrary, there are regions with good performance and regular performance along the Pacific, Andes, and Amazon. The Andes 6 (4 SL events), Amazon 1 (6 SL events), and Amazon 3 (12 SL events) regions were the ones that had the least number of events for calibration and validation. The other regions included more than 10 events (Figure 8), highlighting the Andes 2 (98 SL events), Andes 4 (65 SL events), Amazon 2 (54 SL events) and Pacific 1 (46 SL events) regions.

### 3.3 Effect of Antecedent Conditions

It is known that the antecedent conditions of the terrain play an important role in the occurrence of landslides, and especially in their magnitude. This is the reason why this scenario was analyzed, and included the separation of rain events that only

**Table 3.** Number of SL events and best thresholds for one and two variables for each region (Th: threshold, SL: number of landslides per region, Cal: Calibration, Val: Validation)

| Region | SL total | SL Cal | SL Val | Best Th - 1 variable | TSS | Best Th - 2 variables | TSS |
|--------|----------|--------|--------|----------------------|-----|-----------------------|-----|
| Pacific 1 | 46 | 43 | 3 | $I_{max}$ | 0.68 | $I_{max} - D$ | 0.71 |
| Pacific 2 | 27 | 20 | 7 | $I_{mean}$ | 0.61 | $I_{mean} - D$ | 0.61 |
| Andes 1 | 34 | 28 | 6 | $I_{mean}$ | 0.43 | $I_{mean} - D$ | 0.44 |
| Andes 2 | 98 | 83 | 15 | $E$ and $I_{mean}$ | 0.58 | $I_{max} - D$ | 0.64 |
| Andes 3 | 17 | 10 | 7 | $I_{max}$ | 0.92 | $I_{max} - D$ | 0.91 |
| Andes 4 | 65 | 54 | 11 | $E$ | 0.51 | $I_{mean} - D$ | 0.52 |
| Andes 5 | 14 | 7 | 7 | $E$ | 0.67 | $I_{mean} - D$ and $E - D$ | 0.66 |
| Andes 6 | 4 | 3 | 1 | $D$ | 0.68 | $E - D$ | 0.65 |
| Amazon 1 | 6 | 6 | - | $I_{mean}$ | 0.74 | $I_{mean} - D$ | 0.77 |
| Amazon 2 | 54 | 41 | 13 | $E$ | 0.57 | $E - D$ | 0.58 |
| Amazon 3 | 12 | 10 | 2 | $E$ | 0.68 | $I_{mean} - D$ and $I_{max} - D$ | 0.73 |

consider the rate of rain until a day before the day of landslide occurrence (Table 1). It is observed that, in the calibration phase, the antecedent event scenario obtained lower returns than the integer event scenario. However, in the validation stage for the year 2020, it was observed that, for some thresholds in isolation, their performance was higher; for example, for the Pacific 1 region, the $I_{max}$ and $I_{mean}$ thresholds obtained higher performances than the entire event scenario (including the rainfall rate of the mm event day). This means that in the days prior to the day of occurrence, there was a day with intense rain greater than that on the day of occurrence, and this allows the separation of that event as a triggering event, in addition to altering the average rainfall rate associated with said event.

### 3.4 Evaluation of threshold performance

Validation was carried out for the events that occurred in 2020 by simulating the operability of the calibrated thresholds in a regional alert system. The Amazon 1 region did not contemplate landslide events for that year so it did not enter this assessment. The validation shows that, in most regions and thresholds, there was a clear magnitude decrease (Tables 1 and 2). For example, the $I_{max}$ threshold, which obtained the best performance in calibration, decreased for this period, except for the Andes 4, Andes 6, and Amazon 3 regions, which improved in this validation; this means that the threshold allowed for the separation of the rainfall events of 2020 better than expected in calibration.

The variable $D$ was confirmed to be, by itself, a bad threshold separator for the separation of triggering rain events from those that are not triggering. Even with negative performances (Pacific 1, Andes 1, and Amazon 3), this negativity was associated with the sensitivity (correct prediction of landslides) of the model for these regions, which was 0; i.e., the estimated threshold in the calibration was not able to separate the rainfall events. However, this variable shows that we can associate landslides with continuous rainfall events with an antecedent duration of 8 days.

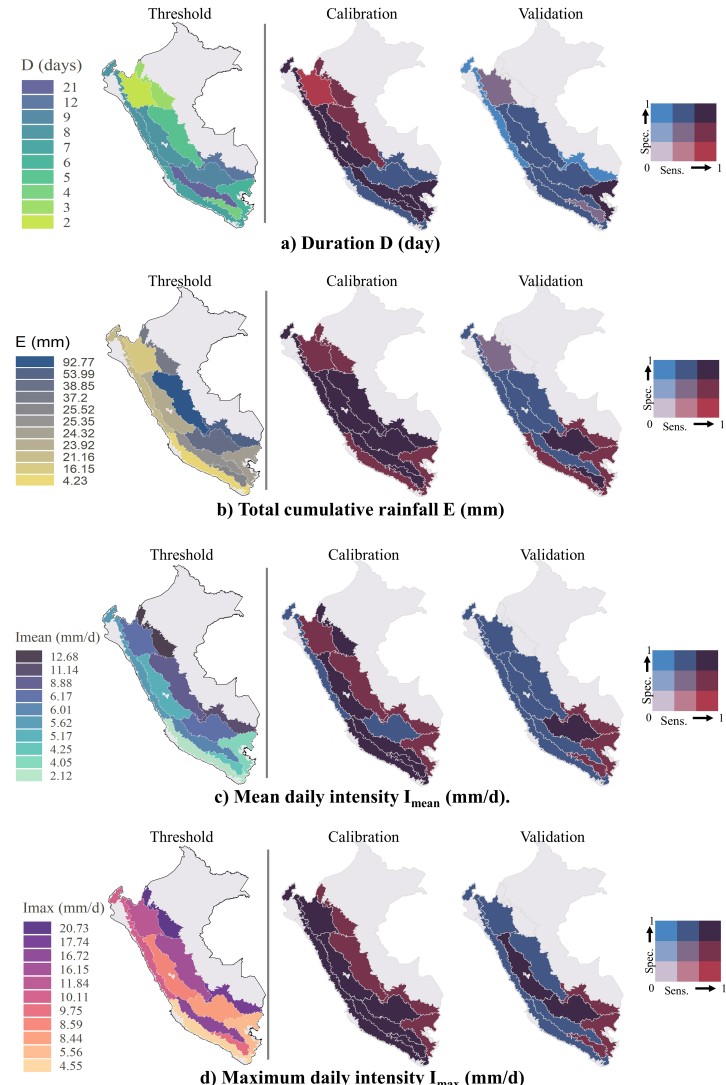

**Figure 7.** The first column shows the spatial distribution of Rainfall thresholds for independent variables magnitude for Peru: (a) day $D$ (days), (b) total cumulative rainfall $E$ (mm), (c) mean daily intensity $I_{mean}$ (mm/day) and (d) maximum daily intensity $I_{max}$ (mm/day). The second and third columns show the bivariate maps indicating the spatial distribution of the $sensitivity$ (probability of correctly predicting landslide triggering rainfall events) and $specificity$ (probability of correctly predicting non-triggering rainfall events from landslide) of the thresholds for calibration and validation.

Regarding the variability of the thresholds (Figure 6), we can explain it mainly to the rainfall climatology in Peru. It can be seen that the magnitudes have a relationship concerning the spatial distribution of rainfall in Peru, that is, low thresholds related to rainfall of lesser magnitude in the arid zones in the western part of Peru (Pacific region), thresholds intermediates related to

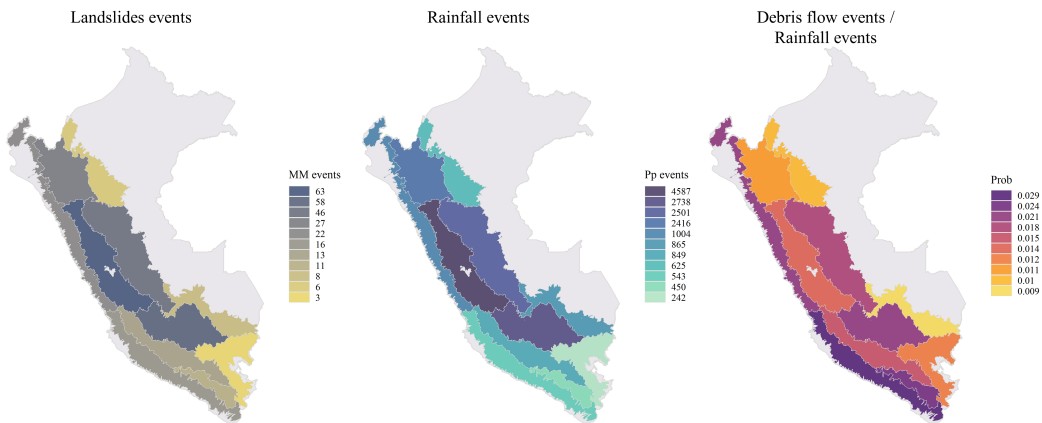

**Figure 8.** Spatial distribution at regional scale of number of landslides events (left), number of rainfall events (middle) and a probability (right) of landslides triggering rainfall event.

the increase in the magnitude of rainfall in the middle part or mountainous region (Andes region) and the highest thresholds related to wet regions (Amazon region). However, the Andes 1, Andes 3 and Andes 6 regions do not have this relationship, so this discussion is not conclusive and is considered to be related to limited data, so it is suggested that this variability be discussed in future research that includes more shallow landslides events data.

Regarding the validation period, 61 events were used in total, resulting in the TSS statistic being more sensitive, mainly due to the increased sensitivity of the model (i.e., the probability of correctly predicting landslide-triggering rainfall events), while specificity remained approximately the same (i.e., the probability of correctly predicting non-landslide triggering rainfall events). This effect points to the importance of obtaining wide and robust inventories of landslides.

The calibration/validation methodology, based on taking one year of observations for the validation set, which was used in other research works (e.g., Kirschbaum et al., 2015b; Dikshit et al., 2019), is quite short and there is the risk of overinterpretation. It is therefore highly recommended for future research to expand the dataset and explore other calibration/validation methods, for example, a random selection of the differentiated data set for the calibration and validation (e.g., 70% for calibration and 30% for validation) (e.g., Brunetti et al., 2021; Gariano et al., 2020).

## 4 Discussions

In this research, rainfall thresholds were determined that allow for the separation of triggering and non-triggering rainfall events for shallow landslide occurrence in two scenarios based on the variables of rainfall events associated with observed landslides. This type of analysis has already been objectively developed in previous studies (Peruccacci et al., 2017, 2012; Segoni et al., 2014; Rosi et al., 2012; Leonarduzzi et al., 2017; Uwihirwe et al., 2020; Abraham et al., 2019). This work is the

first approximation of regional thresholds on a national scale in Peru, and will serve as a starting point and reference for the continued development of this type of research in Peru.

The estimated thresholds of independent variables are show in Table 1 and curve thresholds are show in Table 2. The thresholds of independent variables show that the thresholds with the best performance were E for the individual variables of rainfall events and $I_{mean} - D$ for curve thresholds. The variable that had the lowest performance was the duration of the event, $D$, so it should not be used independently, but combined with other event variables. However, it allows us to associate landslide events with the background rain conditions of the previous 8 days, an association that can be used for future research.

Concerning the thresholds of two variables or curves, the TSS had a slight improvement, all exceeding 0.5 in the calibration of the $I_{mean} - D$ threshold (the threshold with the best performance for curved thresholds), except for Andes 1. This efficiency statistics, based on the optimization model has an approach based on maximizing the TSS, through which a high detection rate of landslides (sensitivity) is sought, maintaining, as far as possible, a low rate of detection of false positives (specificity). However, it was observed that to seek this optimization, the detection of landslides is sacrificed (giving false negatives), though false alarms are reduced, and this is a dilemma in terms of alert systems, but TSS is a good balance between landslides detection and false alarms.

The Pacific 1 region is constantly impacted by shallow landslides and also contains most of the cities with the highest population density in Peru, so their evaluation is highly relevant. In this region, it was observed that the $I_{max}$ (TSS = 0.68) and $I_{max} - D$ (TSS = 0. 71) were the best thresholds for the entire event scenario, which indicates that the catchments in this region are highly susceptible to events of maximum intensity. While the $I_{max}$ (TSS = 0.65) and $I_{mean} - D$ (TSS = 0.68) thresholds were the best thresholds for the antecedent event scenario. The $I_{max}$ variable had the best performance, which suggests that high-intensity rains have a high conditioning impact on landslide development. Regarding the validation performances in the antecedent conditions scenario were higher in the calibration performances, it may be because the validation set is too small.

Regionalization was necessary given the high climatic variability in Peru, evidenced by the differences in magnitude between the thresholds. This regionalization helped us to observe the regions of Peru where there is greater landslide occurrence and response to this type of daily threshold. For example, we observed that the Andes 2 region (the region with the highest number of events recorded in recent years) had a better response for the $I_{max}$ threshold for both calibration and validation. Hirschberg et al. (2021) found that 25 events are enough to limit the uncertainties in the ID threshold parameters to $\pm 30\%$ in his study, based on this, it is observed that there are several regions (Andes 3, 5, 6 and Amazon 1, 2) that do not exceed that quantity, so these regions have a greater source of uncertainty due to the quantity of the data. A summary of the number of shallow landslide events used for the research and the thresholds with best performances per region is presented in Table 3.

The evaluation of the performance of the thresholds was carried out through validation with the events of 2020. However, it was observed that the performances decreased, which may be due to the fact that, in the year 2020, there were no extreme rainfall events as in other years, and the number of landslides was lower than in other years. Even the Amazon 1 region had no record of activation events, thus we can state that the low performance was because the thresholds do not represent landslide events with low-impact magnitude, and this associated with one of the focuses of the model, which is to reduce the rate of false alarms.

There are still many different sources of limitations on studies at the regional level in the field of landslides and their interrelation with rainfall as a triggering agent in Peru. The main source of uncertainty in this study was the unreliability of the available databases used, which resulted in the following limitations: (i) rainfall of PISCOpd_Op by the spatio-temporal resolution with a grid of 10 km and a daily time scale; (ii) the basins or units of analysis, which covered several streams, torrents and small basins; (iii) landslides registered, since the objective of the study did not focus on a review of the record of events, although a global analysis of the databases was carried out; (iv) the small number of events recorded in the landslide historical series must also be taken into account; and (v) the climatic regions, due to the great landslide spatial variability of descriptor variables studied in this research. These described factors are limitations with regard to the determination of thresholds and create uncertainties in the generation of regional thresholds that are translated into the performance indices used for the evaluation of thresholds.

## 5 Conclusions

This study is the first approximation of the regional rainfall thresholds that trigger landslides in Peru. It was conducted to estimate and analyze the interrelation between rainfall and its landslide trigger effect in 11 rainfall regions in Peru using an empirical--statistical method. The advantage of this study is the use of landslides datasets available at the national scale to objectively determine and compare rainfall thresholds. Daily gridded rainfall data and landslide records allowed us to estimate landslide-triggering rainfall events and thus determine the properties of triggering and non-triggering rainfall events at susceptible sites, using them to ascertain rainfall thresholds for the activation of shallow landslides triggered by rainfall and to validate their performance. Our main conclusions are:

a. The generation of thresholds using the empirical–statistical method and calibrations based on minimum radial distance and maximum true skill statistics (TSS) were successful in defining rainfall thresholds for landslides. The best predictive performance was obtained using the mean intensity-duration ($I_{mean} - D$) threshold curve, followed by the accumulated daily intensity $E$. The duration of the event independently has very low predictive power.

b. The performances of the calibrated thresholds had a high differentiation between regions. This performances difference is associated with the high variability of rainfall events and their properties in each region, where it is observed that the best performances occur in areas where it is easier to separate rainfall events that trigger and non-trigger shallow landslides, which is reflected in high performances (Andes 3, Amazon 1, Amazon 3 and Pacific 1 regions). However, in other regions, this separation between rainfall events is more complex to carry out, since there are more rainfall events with high magnitudes that do not trigger landslides but that exceed the thresholds, reflecting in lower performances (Andes 1, Andes 4 and Amazon 2). Thus, we could assume that in these regions there is a greater incidence of lithology and geology in the occurrence of SL than just the rains.

c. Through the observed PISCOpd_Op and landslides databases, it is possible to generate daily rainfall thresholds for shallow landslide occurrence. However, the uncertainties associated with these databases are the main source of uncertainty

for the thresholds. The few landslides recorded made the validation performance highly sensitive to the few data (i.e., a single event could lead to a high or low value of the performance statistics).

The results of this work demonstrate the potential of rainfall thresholds based on the characteristics of rainfall events associated with landslides for implementation in landslide monitoring in Peru. Future work should focus on three main perspectives based on the limitations and sources of uncertainty: i) improvement in the spatio-temporal resolution of gridded rainfall; ii) improvement in the spatial discretization of regions where the greatest number of landslides take place, which is dependent firstly on improving the spatio-temporal resolution of rainfall; and iii) the assimilation of landslide databases to improve the certainty of the thresholds and reduce their sensitivity.

*Code and data availability.* The source code with an example data set is available from GitHub (https://github.com/caemillan/Rainfall_thresholds_for_shallow_landslide.git).

*Author contributions.* CM conducted the analysis and interpreted the results. WL and CM conceived the research and prepared the paper.

*Competing interests.* The authors declare that they have no conflict of interest.

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
