# Peer review of "Rainfall thresholds estimation for shallow landslides in Peru from gridded daily data"

_Natural Hazards and Earth System Sciences, 2022_

## Referee Comment (RC1)

**GENERAL COMMENTS**

Millan-Arancibia and Lavado-Casimiro describe how they determine regional rainfall thresholds for landslides in Peru at the national scale. They use recently developed methods to objectively assess these thresholds, which makes the study interesting more from a technical than from a scientific perspective for those who aim at implementing early-warning systems. The study is a bit hard to read and seems unorganized in some parts. As a consequence, parts of the methods, results and conclusions were not clear to me. I have few general comments and more specific ones below, which should be addressed before publication in NHESS.

1. There are quite many specifications and clarifications needed in order to make the methods they used unambiguous and reproducible. This also resulted in quite a long list of specific comments below.

2. Some paragraphs seem unnecessary wordy or seem like a random list of unrelated statements, which makes it difficult to follow. For example, L. 177 "TSS is more objective than simple random estimate", it could be explained what makes TSS objective (e.g. balancing TPR and FPR). Some of these arguments are in the text but unorganized and unclear. I think the authors will easily identify such paragraphs themselves when editing. See also comments below.

3. I miss mainly two discussion points. One is the spatial variability of thresholds and the origin of this. Can it be explained with climatology/lithology or is it related to the quality of the data set? See also comments to Figure 7. The second point is related to how calibration/validation is performed, there is almost no discussion about that. I appreciate that this important step is taken and I understand that the dataset is new and short. However, I think it should be stated more clearly that a validation set of one year is quite short and there is a risk of overinterpreting. I suggest to at least discuss other possible validation techniques than splitting years, and flag that as a topic for future research.

4. There are some results and conclusions that are not clear or surprising to me, which should be checked. For example, I would expect Imean-D and E-D thresholds to result in the same performance, but this is not the case here. See comments below.

**SPECIFIC COMMENTS**

L. 24: Citation needed for the original cause and the different processes leading to saturation

L 27: (e.g. Prenner…)

L. 31: rainfall thresholds

L. 35: time

L. 31: The literature you cite only consider statistical methods. Berti et al. (2020) or Tang et al. (2019) are examples for thresholds based of physically-based modelling. Please also change "physical bases" to "physically-based models"

L. 37: in the way it's written it makes one think that the difference between the global and national rainfall thresholds is that one is based on antecedent precip and the other on empirical-statistical approaches. Please rephrase. Also, if you use "antecedent", does it have the same meaning as in L. 29? Antecedent conditions can refer to the conditions prior to the triggering rainfall or prior to the exact time of landslide occurrence. Please specify and use consistently.

L. 45: I think this section is to justify the methods used. Given the uncertainties in the rainfall product that you mention later in the ms one could ask why you're not using physically-based modelling, which consider the actual mechanisms causing landslides, to back-calculate rainfall thresholds. Hence, I would also mention the challenged accompanied with such models: mainly the many high-quality input data such as soil information that is needed, which is associated with high uncertainties, too.

L. 56: maximum at what scale? Daily, annual?

L. 60: gridded

L. 80: Just out of curiosity. It's funny enough that the precipitation dataset is named after Peru's national liquor. Is PISCOpd_Op actually the abbreviation of something?

L. 84: Can you give some information on the number of rain gauges or the average distance? Maybe even add them to the map in Figure 2 if you have such a map.

L. 85: What do you mean by "multipliers that are based on monthly climatology"?

Table 1: I'm not sure this table is so important. To me, only the spatial resolution and the time period is of relevance. But why compare these two datasets if you only use one of them?

L. 92-93: these two sentences can be simplified, now it is confusing. So SLIP covers the period 2018-2020 but you have greater certainty for 2019 and 2020?

L. 101: Figure 3

L. 88-101: I don't understand how the two landslide databases were combined. They time periods do not overlap and Figure 3 only starts at 2019. If one event was excluded it should be 382 events in total. So which was your study period?

Figure 3: What is this rainfall? One grid cell? Which location are we looking at? And the colour is one rainfall event? Please specify in the caption and add labels a) and b= to subplots.

L. 103: Since you describe the sequence of your methods here, Figure 1 would fit here. And describe the steps in the text and refer to the figure.

L. 116: How can the PISCO report Pr>0 and the station Pr=0 if Pisco is interpolated from the stations?

L. 118: How were rainfall events defined? Are two events independent if they are separated by at least one non-rainy day?

L. 131: events

L. 134: I think that E-D and Imean-D should result in the same thresholds, only that b(E-D) = b(Imean-D)+1. That's what I get when substituting Imean with E/D. So there is no point in comparing both thresholds. This said, I'm surprised that by the number is table 3. Either I'm misunderstanding something or something went wrong here. Please clarify.

L. 135: a and b are scale and shape parameters, but in the log-log space they become intersection and slope of the linear threshold

Figure 4: These box plots are nice but it's not clear from the text why you show them. Is it to show that the two can be separated well? Considering the methods you use, it would be nice to see some AUC curves instead which would also help you in explaining the methods

L. 146: Max precip at what time scale? And what is the motivation for using this for regionalization and one of the other indices?

L. 158: Please consider rewriting or reorganizing this section. The information to certain steps are spread across the entire section, for example, how the dataset was split into calibration and validation data sets.

L. 179-182: This will be confusing for many readers. You have two definitions for TSS, and two for sensitivity. Please be consistent and avoid introducing alternative definitions if they actually mean the same. Also, the TSS itself doesn't seek to maximize TPR and 1-FPR, but you do so by choosing a threshold that maximizes TSS.

L. 185: Please be more specific. It's not clear what you did using ROC, TPR, FPR. Which is the "most widely used technique"? Did you choose some variables with large AUC and dropped the others. If so, what was the threshold AUC. Or did you define thresholds by maximizing TSS? There are many possible

L. 192: It's not clear to me how exactly the validation was performed. Was the performance of the validation data set calculated for the thresholds determined with the calibration data set or was a new threshold determined for the validation data set to see if the performance is similar?

L. 196: The values of 0.4 and 0.7 seem somewhat random. Could you elaborate a bit on the meaning of these values? Are these values commonly used or why is this classification needed?

L. 205-214: I'm surprised that Imean-D and E-D don't have the same performances. See comment L. 134.

Table 2: "D (days)". Is this the full data set, calibration or validation? How many events per region? The same for Table 3.

L. 270: do you mean first in Peru? Please specify.

L. 277: Table 3

L. 280: Yes, landslide detection is sacrificed but false alarms are reduced. There are various scores one could chose depending on if you want to give more weight to the detection or false alarms. But you chose TSS because it's a good balance between the two.

L. 283: What is a high-impact stream?

L. 284: what do you mean by constant landslide occurrence?

L. 284: Imax-D-D?

L. 285: di xyou mean entire event?

L. 286: is background condition scenario the antecedent condition scenario?

L. 286-290: I can't follow. If you're the validation results are better than the calibration, then maybe your validation set is too small. I don't see how you can conclude the importance of antecedent conditions from this. Also, the sentence "in the validation stage…showed growth in calibration performance" is confusing

L. 296: The absence of extreme events does not imply poorer threshold performance. An option would be to do calibration/validation on more data splits.

L. 298: "the number of landslides was lower than in other years" but the only reliable year you can compare with is 2019, right?

L. 313: Again, you mean first in Peru, right? Please specify.

L. 315: Well, you cannot compute empirical-statistical thresholds without landslide observations so this is not really an advantage. An advantage is that you have used datasets available at the national scale to objectively determine and compare rainfall thresholds.

L. 318: it is still not entirely clear to me what process we are talking about. Here you say shallow landslide and earlier you mention streams and debris flow. Is it a mix of processes? Please add some information on this in the dataset description and clearly define what collection of processes you refer to when using "landslide" throughout the ms.

L. 324: More interesting would be why the performances can be so different. Can you say something about that?

L. 329: high sensitivity to what?

Figure 7: Is there a reason for showing sensitivity/specificity? Wouldn't it be easier to interpret if you would just colour according to TSS?
This figure is very interesting and shows high spatial variability in the thresholds. Can you say something about this variability? E.g. is the threshold higher in wet regions? See e.g. Leonarduzzi et al. (2017) Figure 7 or Marc et al. (2019).

Marc, O., Gosset, M., Saito, H., Uchida, T., Malet, J.P., 2019. Spatial Patterns of Storm-Induced Landslides and Their Relation to Rainfall Anomaly Maps. Geophys. Res. Lett. 167–177. https://doi.org/10.1029/2019GL083173

---

## Author Comment (AC1)

**RESPONSE TO GENERAL COMMENTS**

Millan-Arancibia and Lavado-Casimiro describe how they determine regional rainfall thresholds for landslides in Peru at the national scale. They use recently developed methods to objectively assess these thresholds, which makes the study interesting morefrom a technical than from a scientific perspective for those who aim at implementing early-warning systems. The study is a bit hard to read and seems unorganized in some parts. As a consequence, parts of the methods, results and conclusions were not clear tome. I have a few general comments and more specific ones below, which should be addressed before publication in NHESS.

**Comment response:** Thank you very much for your review, in the new version of the mn we have tried to make it not a bit difficult to read and also not seems unorganized, considering all your comments. Additionally, this document is highly important for the scientific community related to landslides in Peru since this type of work has not been developed in Peru, which, in addition, faces the limited availability of data compared to other countries. Lastly, other investigations also faced similar difficulties (e.g., Kirschbaum et al., 2015; Abraham et al., 2019).

1.  There are quite many specifications and clarifications needed in order to make the methods they used unambiguous and reproducible. This also resulted in quite a long list of specific comments below.

    **Comment response:** Thanks for the comment. All your comments and the list of specific observations have been taken into account and included in the new version of the mn.

2.  Some paragraphs seem unnecessary wordy or seem like a random list of unrelated statements, which makes it difficult to follow. For example, in L. 177 "TSS is more objective than simple random estimate", it could be explained what makes TSS objective (e.g. balancing TPR and FPR). Some of these arguments are in the text but unorganized and unclear. I think the authors will easily identify such paragraphs themselves when editing. See also comments below.

    **Comment response:** Thanks for the observation. All your comments and the list of specific observations have been taken into account and included in the new version of the mn. We have made an exhaustive revision of the mn and we have identified some paragraphs and we have organized them with greater clarity to avoid their difficult reading.

3.  I miss mainly two discussion points. One is the spatial variability of thresholds and the origin of this. Can it be explained with climatology/lithology or is it related to the quality of the data set? See also comments to Figure 7. The second point is related to how calibration/validation is performed, there is almost no discussion about that. I appreciate that this important step is taken and I understand that the dataset is new and short. However, I think it should be stated more clearly that a validation set of one year is quite short and there is a risk of overinterpreting. I suggest at least to discuss other possible validation techniques than splitting years, and flag that as a topic for future research.

    **Comment response:** Thanks for the observation. We have taken into account your observations and recommendations and have included them in the discussions of the new version of the mn. Regarding the first point of discussion:

    "Regarding the variability of the thresholds, we can explain it mainly to the rainfall climatology in Peru. It can be seen that the magnitudes have a relationship with respect to the spatial distribution of rainfall in Peru, that is, low thresholds related to rainfall of lesser magnitude in the arid zones in the western part of Peru (Pacific region), thresholds

intermediates related to the increase in the magnitude of rainfall in the middle part or mountainous region (Andes region) and the highest thresholds related to wet regions (Amazon region). However, the Andes 1, Andes 3 and Andes 6 regions do not have this relationship, so this discussion is not conclusive and is considered to be related to limited data, so it is suggested that this variability be discussed in future research that include more shallow landslides events data."

Just to comment, that the lithology in Peru is still highly general and we hope in the future to do exercises with lithological data (e.g., soil tests) that we are developing at small basins level.

About the second point, regarding calibration/validation we have added your observation and we have discussed about it, as you can see below:

"The calibration/validation methodology, based on take one year of observations for validation set, which was used in other research works (e.g., Dikshit et al., 2019; Kirschbaum et al., 2015), is quite short and there is the risk of overinterpretation. It is therefore highly recommended for future research to expand the dataset and explore other calibration/validation methods, for example, a random selection of the differentiated data set for the calibration and validation (e.g., 70% for calibration and 30% for validation) (Brunetti et al., 2021; Gariano et al., 2020)."

In addition, in our future research we hope to advance in these limitations in Peru, for example, our perspective is to expand the database, for which we are working with INDECI (entity in charge of the attention of the population when landslides occur) for future studies that include greater data extension.

4. There are some results and conclusions that are not clear or surprising to me, which should be checked. For example, I would expect Imean-D and E-D thresholds to result in the same performance, but this is not the case here. See comments below.

   **Comment response:** Thanks for observation. We have taken into account your comment. For better understand, according to the way we have defined the variables for a dataset, Imean, that is affected by D, does not have the same distribution as E. For example, two events with the same E (e.g. E=10), can have different D (e.g. D equal to 2 and 4 days), therefore, the Imean of both resulting events are different (Imean equal to 5 and 2.5 respectively), so the threshold could not be defined as the division of both. A more specific example for a example dataset is shown in the specific comments below.

**RESPONSE TO SPECIFIC COMMENTS**

L. 24: Citation needed for the original cause and the different processes leading tosaturation

**Comment response:** Thanks for the observation. The citation is: Lynn Highland. 2006. Landslide Types and Processes. USGS Fact Sheet 2004–3072. But it was removed for better understand according the general comments.

L. 27: (e.g. Prenner…)

**Comment response:** Thanks for the observation. It was edited in the new version of the mn.

L. 31: rainfall thresholds

**Comment response:** Thanks for the observation. It was edited.

L. 35: time

**Comment response:** Thanks for the observation. It was edited.

L. 31: The literature you cite only considers statistical methods. Berti et al. (2020) and Tang et al. (2019) are examples of thresholds based on physically-based modelling. Please also change "physical bases" to "physically-based models"

**Comment response:** Thanks for the observation. It was added the citation examples and edited "physical bases" to "physically-based models". Additionally, we have recently instrumented some basins to collect more accurate data for future research, where we could explore physically-based models.

L. 37: in the way it's written it makes one think that the difference between the global and national rainfall thresholds is that one is based on antecedent precip and the other on empirical-statistical approaches. Please rephrase. Also, if you use "antecedent", does it have the same meaning as in L. 29? Antecedent conditions can refer to the conditions prior to the triggering rainfall or prior to the exact time of landslide occurrence. Please specify and use consistently.

**Comment response:** Thanks for the observation. The text has been rephrased in order to clarify the main idea, as you can see below.

"For example, there is been developed empirical–statistical approach to the estimation of global thresholds (Caine, 1980; Guzzetti et al., 2008; Kirschbaum and Stanley, 2018), and national thresholds (Leonarduzzi et al., 2017; Peruccacci et al., 2017a; Uwihirwe et al., 2020)."

L. 45: I think this section is to justify the methods used. Given the uncertainties in the rainfall product that you mention later in the ms one could ask why you're not using physically-based modelling, which considers the actual mechanisms causing landslides, to back-calculate rainfall thresholds. Hence, I would also mention the challenges accompanied with such models: mainly the many high-quality input data such as soil information that is needed, which is associated with high uncertainties, too.

**Comment response:** Thanks for the observation. It was edited, as you can see below.

"This empirical approach is widely applied because its analysis and implementation do not require the constant monitoring of the other physical variables on which other types of most robust models are based (e.g., physically-based models), and this drawback of the robust models is the main advantage of empirical approaches and its applicability over large areas (Rosi et al., 2012). Another advantage for its application is that it is not subject to the challenges accompanied with other models, mainly the many high-quality input data, such as soil information that is needed, which is associated with high uncertainties too."

To comment, we are recently developing studies on a local scale with less uncertainties that we will use to define rainfall thresholds at local scale (Asencios Astorayme, 2020a, b). https://repositorio.senamhi.gob.pe/handle/20.500.12542/478 https://repositorio.senamhi.gob.pe/handle/20.500.12542/476

L. 56: maximum at what scale? Daily, annual?

**Comment response:** Thanks for the observation. It´s daily scale. It was edited.

L. 60: gridded

**Comment response:** Thanks for the observation. It was edited.

L. 80: Just out of curiosity. It's funny enough that the precipitation dataset is named after Peru's national liquor. Is PISCOpd_Op actually the abbreviation of something?

**Comment response:** Thanks for the observation. Yeah, the name helped us a lot as a hydrometeorological service to be able to spread the information in a fun way. The PISCO is derived from: **P**eruvian **I**nterpolated data of the **S**ENAMHI's **C**limatological and Hydrological **O**bservations. PISCO is a base name of different products of SENAMHI, i.e., PISCOpd_Op is derived from **PISCO P**recipitation-**D**aily-**Op**erative Gridded data. It was edited for better understanding, as you can see below.

L. 84: Can you give some information on the number of rain gauges or the average distance? Maybe even add them to the map in Figure 2 if you have such a map.

**Comment response:** Thanks for the observation. For the PISCOpd_Op purpose, we use 416 rain gauges and them were added to Fig 1 (before Fig 2).

[Figure]

L. 85: What do you mean by "multipliers that are based on monthly climatology"?

**Comment response:** Thanks for the comment. These multipliers are the ratio between the value of the monthly background grid at location x (extracted from PISCOp monthly climatology) and the value of the monthly back- ground grid at the gauge location for every gauge (derived from rain gauges) to create a set of multipliers from the gauges to the given grid cell. For more information about genre Interpolation Method is shown in: van Osnabrugge, B., Weerts, A. H., & Uijlenhoet, R. (2017). genRE: A method to extend gridded precipitation climatology data sets

in near real-time for hydrological forecasting purposes. Water Resources Research, 53, 9284–9303. https://doi.org/10.1002/ 2017WR021201.

Table 1: I'm not sure this table is so important. To me, only the spatial resolution and the time period are of relevance. But why compare these two datasets if you only use one of them?

**Comment response:** In consideration of the observation, we decided to remove the table and show only the relevant information (i.e., spatial resolution and the time resolution).

L. 92-93: these two sentences can be simplified, now it is confusing. So SLIP covers the period 2018-2020 but do you have greater certainty for 2019 and 2020?

**Comment response:** Thanks for the observation. The SLIP covers the period 2014-2020, it was corrected, and we have more certainty from 2019-2020 just because we were more data and number of events these last years. It was edited, as you can see below.

SLIP was implemented in January 2019 and has 330 records from the 2014–2020 period. Therefore, there is a greater degree of certainty regarding the number of events recorded in recent years.

L. 101: Figure 3

**Comment response:** Thanks for the observation. It was edited.

L. 88-101: I don't understand how the two landslide databases were combined. The time periods do not overlap and Figure 3 only starts in 2019. If one event was excluded it should be 382 events in total. So which was your study period?

**Comment response:** Thanks for the observation. According previous comment, the period was 2007-2020. The number of events was edited. The figure it's just an extracted period for show how we define an event.

Figure 3: What is this rainfall? One grid cell? Which location are we looking at? And the colour is one rainfall event? Please specify in the caption and add labels a) and b= to subplots.

**Comment response:** Thanks for the observation, we have taken into account your comment and the figure has been modified. It's a daily rainfall data for one basin (from GEOGloWS discretization, fig1) where occurred a landslides event. The purpose of the figure was to show how its defined rainfall events (each color it´s a rainfall event). The figure it's just an extracted period for show how we define an event. It was edited, as you can see below.

[Figure]

**Figure 3.** a) Extract from the precipitation time series (rainy period 2019) for an example basin, where the estimated rainfall events are observed (each color is a rainfall event, the lead-colored event 0 is the non-rainy days). b) An example of a rain event associated with the occurrence of a landslide, in this case the rain event No. 93, where the variables analyzed for the estimation of thresholds are shown: the maximum daily intensity $I_{max}$ (mm/day), the accumulated precipitation $E$ (mm), the duration $D$ (day), and the mean daily intensity $I_{mean} = E/D$ (mm/day).

L. 103: Since you describe the sequence of your methods here, Figure 1 would fit here. And describe the steps in the text and refer to the figure.

Comment response: Thanks for the observation. It was edited, as you can see on manuscript edited, moreover we put Regionalization subsection before the Rainfall threshold model subsection because we think it help to manuscript better understand.

L. 116: How can the PISCO report Pr>0 and the station Pr=0 if Pisco is interpolated from the stations?

Comment response: Thanks for the observation. The principal reasons for this, is because in the interpolation method it's affected by monthly climatology. Therefore, it is not an exact interpolation, but rather an approximate one, since it tries to represent gridded data at the national scale. Another comment, we are developing another rainfall products that have the purpose of improve the representativeness of rainfall products where there are no terrain data based on novel methodologies with which we think to include them in future research about landslide thresholds. Additionally, the installation of radars and more rain gauges is planned in Peru, which will be assimilated in future rainfall products.

L. 118: How were rainfall events defined? Are two events independent if they are separated by at least one non-rainy day?

Comment response: Thanks for the observation. L 109: "For this work, we define an independent rainfall event as a series of consecutive rainy days where it has rained above a minimum rainfall threshold (Figure 3)".

L. 131: events

Comment response: Thanks for the observation. It was edited.

L. 134: I think that E-D and Imean-D should result in the same thresholds, only that b(E- D) = b(Imean-D)+1. That's what I get when substituting Imean with E/D. So there is no point in comparing both thresholds. This said I'm surprised by the numbers in table 3. Either I'm misunderstanding something or something went wrong here. Please clarify.

**Comment response:** Thanks for observation. We have taken into account your comment. For better understand, according to the way we have defined the variables for a dataset, Imean, that is affected by D, does not have the same distribution as E. For example, two events with the same E (e.g. E=10), can have different D (e.g. D equal to 2 and 4 days), therefore, the Imean of both resulting events are different (Imean equal to 5 and 2.5 respectively), so the threshold could not be defined as the division of both. The Fig. X1 shows what is mentioned for an example dataset, where it is observed that E and Imean have different density distributions and therefore their predictive potentials also change (i.e., the thresholds do not have the same Imean relationship =E/D).

[Figure]

Fig. X1: Density plot of the variables E (a), Imean (b) and D (c) for the same data set, where it is observed that the distributions of the variables E and Imean are different.

L. 135: a and b are scale and shape parameters, but in the log-log space they become the intersection and slope of the linear threshold.

**Comment response:** Thanks for the observation. It was edited as you can see below:

"… a and b are the scale and shape parameters of the curve (while for log-log space a is the intersection parameter and b denotes the slope of the linear threshold)".

Figure 4: These box plots are nice but it's not clear from the text why you show them. Is it to show that the two can be separated well? Considering the methods, you use; it would benice to see some AUC curves instead which would also help you in explaining the methods

**Comment response:** Thanks a lot for the observation. Indeed, the purpose is to show the ability to separated variables, before determining a threshold, and how it change for each region. The AUC would not help much at first, additionally, this way of showing the potential of the variables has been used in other publications (Martinović et al., 2018; Leonarduzzi et al., 2017). (Martinović et al., 2018; Leonarduzzi et al., 2017). We have taken into account your observation and the text was edited, as you can see:

"Figure 4. Boxplot of triggering (yellow) and no triggering (blue) total cumulative rainfall E for the eleven regions stablished in this study for Peru. The boxplot graphs include outliers and show the potential predictive for the E variable to separate the rainfall events that trigger/no trigger shallow landslides. Also, the plot shows the region variability of the rainfall events that trigger shallow landslides."

L. 146: Max precip at what time scale? And what is the motivation for using this for regionalization and one of the other indices?

**Comment response:** Thanks for the observation. It is daily scale (it was edited). We use this Max Dayli Precip regionalization for Peru in addition to the covariates of relief (altitude) and climatology (average precipitation), mainly because we associate these maximum daily rainfall events with rainfall triggered landslides. The altitude to maintain an orographic similarity, since, in Peru, and in general in South America, the Andes have a modulating character in the presence of rains. And the average precipitation derived from PISCOp that helped us to establish a similarity of the basins, especially in the transitions in the limits between each region.

Also, we use these maximum rainfall regions because we took as an initial reference the paper from Leonarduzzi et al., 2017 where they use the Maximum Intensity within their regionalization, which was the one that gave the best results in their threshold estimation. Finally, we already had this regionalization of previous studies, which is related to the map of climatic regions of Peru (SENAMHI, https://repositorio.senamhi.gob.pe/handle/20.500.12542/1336).

L. 158: Please consider rewriting or reorganizing this section. The information to certain steps are spread across the entire section, for example, how the dataset was split into calibration and validation data sets.

**Comment response:** Thanks a lot for the observation. I rewrite and reorganize the entire section for better understand, as you can see below.

"2.6 Calibration and validation of thresholds

[revised manuscript text omitted]

L. 179-182: This will be confusing for many readers. You have two definitions for TSS, and two for sensitivity. Please be consistent and avoid introducing alternative definitions if they actually mean the same. Also, the TSS itself doesn't seek to maximize TPR and 1-FPR, but you do so by choosing a threshold that maximizes TSS.

**Comment response:** Thanks a lot for the observation. I avoided the use of double definition for TSS, I simplify the paragraph.

L. 185: Please be more specific. It's not clear what you did using ROC, TPR, FPR. Which is the "most widely used technique"? Did you choose some variables with large AUC and dropped the others? If so, what was the threshold AUC. Or did you define thresholds by maximizing TSS? There are many possible

**Comment response:** Thanks a lot for the observation. I checked the information and simplified the paragraph, as you can see below:

"For thresholds based on rainfall event properties independently (Imax, E, D or Imean), the overall impression of the predictive power was estimated whit the so-called receiver operating characteristic (ROC) curve (Fawcett, 2006), from which the minimum radial distance to the perfect classificatory test (TSS=1, with se=1 and 1-sp=0) was used to select the individual variable threshold (e.g., Uwihirwe et al.; Gariano et al.; Postance et al.) while for the threshold curve (Imax−D,E−D, Imean−D) the scale parameter a and the shape parameter b are simultaneously tuned to maximize the the true skill statistics (TSS) (e.g., Leonarduzzi et al.; Hirschberg et al.). This maximization was automatically calibrated using the shuffled complex evolutionary algorithm (SCEA-UA) (Duan et al., 1993), considering the TSS as the objective function. The methodology was applied for each region within the analysis area, finding different thresholds for each of them."

L. 192: It's not clear to me how exactly the validation was performed. Was the performance of the validation data set calculated for the thresholds determined with the calibration data set or was a new threshold determined for the validation data set to see if the performance is similar?

**Comment response:** Thanks a lot for the observation. This validation process was computed for landslides occurred on 2020 year using the thresholds calibrated to get the metric for this period and compare the capacity of thresholds to separate rainfall events that trigger shallow

landslides.

"Regarding the validation process, it was consisted of evaluating thresholds calibrated (both individual and curve thresholds) using the landslides events recorded in 2020, which represented approximately 20% of the recorded events. This process was carried out for the year 2020, as we wanted to know how the thresholds would perform when they were assimilated into a regional early warning system."

L. 196: The values of 0.4 and 0.7 seem somewhat random. Could you elaborate a bit on the meaning of these values? Are these values commonly used or why is this classification needed?

**Comment response:** Thanks for the observation. Considering your comments, we agree with the observation. The values are not standardized, in addition to the fact that they were not taken into account in the discussion carried out, so we decided to remove the sentence.

L. 205-214: I'm surprised that Imean-D and E-D don't have the same performances. See comment L. 134.

**Comment response:** Thanks for the observation. It was responded in the observation of the L. 134 from the present text.

Table 2: "D (days)". Is this the full data set, calibration or validation? How many events per region? The same for Table 3.

**Comment response:** Thanks for the observation. It was corrected. The tables shown the thresholds estimated with the calibration set. The number of events is specified on the Table 3 of the new version of the manuscript (previously table 4). As you can see below:

**Table 3.** Number of SL events and best thresholds for one and two variables for each region (Th: threshold, SL: number of landslides per region, Cal: Calibration, Val: Validation)

| Region | SL total | SL Cal | SL Val | Best Th - 1 variable | TSS | Best Th - 2 variables | TSS |
|---|---|---|---|---|---|---|---|
| Pacific 1 | 46 | 43 | 3 | $I_{max}$ | 0.68 | $I_{max} - D$ | 0.71 |
| Pacific 2 | 27 | 20 | 7 | $I_{mean}$ | 0.61 | $I_{mean} - D$ | 0.61 |
| Andes 1 | 34 | 28 | 6 | $I_{mean}$ | 0.43 | $I_{mean} - D$ | 0.44 |
| Andes 2 | 98 | 83 | 15 | $E$ and $I_{mean}$ | 0.58 | $I_{max} - D$ | 0.64 |
| Andes 3 | 17 | 10 | 7 | $I_{max}$ | 0.92 | $I_{max} - D$ | 0.91 |
| Andes 4 | 65 | 54 | 11 | $E$ | 0.51 | $I_{mean} - D$ | 0.52 |
| Andes 5 | 14 | 7 | 7 | $E$ | 0.67 | $I_{mean} - D$ and $E - D$ | 0.66 |
| Andes 6 | 4 | 3 | 1 | $D$ | 0.68 | $E - D$ | 0.65 |
| Amazon 1 | 6 | 6 | - | $I_{mean}$ | 0.74 | $I_{mean} - D$ | 0.77 |
| Amazon 2 | 54 | 41 | 13 | $E$ | 0.57 | $E - D$ | 0.58 |
| Amazon 3 | 12 | 10 | 2 | $E$ | 0.68 | $I_{mean} - D$ and $I_{max} - D$ | 0.73 |

L. 270: do you mean first in Peru? Please specify.
**Comment response:** Thanks for the observation. Yes, the first approximation in Peru. It was edited.

L. 277: Table 3
**Comment response:** Thanks for the observation. It was edited.

L. 280: Yes, landslide detection is sacrificed but false alarms are reduced. There are various scores one could chose depending on if you want to give more weight to the detection or false alarms. But you chose TSS because it's a good balance between the two.
**Comment response:** Thanks for the comment. The paragraph was edited, as you can see:

"However, it was observed that to seek this optimization, the detection of landslides is sacrificed (giving false negatives), though false alarms are reduced, and this is a dilemma in terms of alert systems, but TSS is a good balance between landslides detection and false alarms."

L. 283: What is a high-impact stream?

**Comment response:** Thanks for the question. We refer to high-impact stream a basin with a constant occurrence of landslides. But it's a local phrase, so it was removed for better understand.

L. 284: what do you mean by constant landslide occurrence?

**Comment response:** Thanks for the question. The paragraph was simplified, as you can see below:

"The Pacific 1 region is constantly impacted by shallow landslides and also contains most of the cities with the highest population density in Peru, so their evaluation is highly relevant."

L. 284: Imax-D-D?

**Comment response:** Thanks for the observation. Its Imax-D. It was edited.

L. 285: do you mean entire event?

**Comment response:** Thanks for the observation. Yes it´s the entire event. It was edited.

L. 286: is the background condition scenario the antecedent condition scenario?

**Comment response:** Thanks for the observation. Yes it´s the entire e antecedent event scenario. It was edited.

L. 286-290: I can't follow. If you're the validation results are better than the calibration,then maybe your validation set is too small. I don't see how you can conclude the importance of antecedent conditions from this. Also, the sentence "in the validation stage…showed growth in calibration performance" is confusing.

**Comment response:** Thanks a lot for the observation. The paragraph was edited, as you can see:

"The Imax variable had the best performance, which suggests that high-intensity rains have a high conditioning impact on landslide development. Regarding the validation performances in the antecedent conditions scenario were higher in the calibration performances, it may be because the validation set is too small."

L. 296: The absence of extreme events does not imply poorer threshold performance. An option would be to do calibration/validation on more data splits.

**Comment response:** Thanks for the observation. Regarding calibration/validation we have added your observation and we have discussed about it. The paragraph was edited, as you can see:

"The calibration/validation methodology, based on take one year of observations for validation

set, which was used in other research works (e.g., Dikshit et al., 2019; Kirschbaum et al., 2015), is quite short and there is the risk of overinterpretation. It is therefore highly recommended for future research to expand the dataset and explore other calibration/validation methods, for example, a random selection of the differentiated data set for the calibration and validation (e.g., 70% for calibration and 30% for validation) (Brunetti et al., 2021; Gariano et al., 2020)".

In addition, we add the recommendation that taking only one year for validation may be inconclusive due to the little data, so it should be taken into account in future studies and explore more data splits.

L. 298: "the number of landslides was lower than in other years" but the only reliable year you can compare with is 2019, right?

**Comment response:** Thanks for the observation. The calibration was made with landslides occurred before 2020 and validation with landslide occurred in 2020.

The paragraph was edited in the new version of mn as you can see:

"For the calibration, all events occurring before 2020 were selected, representing approximately 70% of the recorded events. Regarding the validation process, it was consisted of evaluating thresholds calibrated using the landslides events recorded in 2020, which represented approximately 30% of the recorded events."

L. 313: Again, you mean first in Peru, right? Please specify.
Comment response: Thanks for the observation. Yes, the first approximation in Peru. It was edited:

"This study is the first approximation of the regional rainfall thresholds that trigger landslides in Peru."

L. 315: Well, you cannot compute empirical-statistical thresholds without landslide observations so this is not really an advantage. An advantage is that you have used datasets available at the national scale to objectively determine and compare rainfallthresholds.
Comment response: Thanks for the observation. This recommend was incorporated, as you can see:
"The advantage of this study is the use of landslides datasets available at the national scale to objectively determine and compare rainfall thresholds".

L. 318: it is still not entirely clear to me what process we are talking about. Here you say shallow landslide and earlier you mention streams and debris flow. Is it a mix of processes? Please add some information on this in the dataset description and clearly define what collection of processes you refer to when using "landslide" throughout the ms.

Comment response: Thanks for the observation. We mention streams only to refer a body of flowing water. Regard the processes, we included debris flow category which are shallow in nature (Naidu et al., 2018) into shallow landslide term. A clarification to this was added to the new version of the mn.

"The second main source of information used for this research was two inventories of observed and collected landslide events: SENAMHI's of Rainfall-Triggered Shallow Landslides Inventory of Peru (SLIP) and NASA's Global Landslide Catalog (GLC) (Kirschbaum et al., 2015a). Both catalogs consider all types of shallow landslides triggered by rainfall that have been reported in the media, in databases of agencies associated with disasters, in scientific reports, and in other available sources. Most of them belong to the debris flow category which are shallow in nature

(Naidu et al., 2018). In this sense, in this study was used shallow landslide (SL) for all types of shallow landslide processes."

L. 324: More interesting would be why the performances can be so different. Can you say something about that?

Comment response: Thanks a lot for the observation. The differentiation of threshold yields for each region responds to the high variability of rainfall events and their properties (see Figure 4 Boxplot and Figure 7 threshold plots) in each region, we explain this topic and add the next conclusion, as you can see below:

"The performances of the calibrated thresholds had a high differentiation between regions. This performances difference is associated with the highly variability of rainfall events and their properties in each region, where it is observed that the best performances occur in areas where it is easier to separate rainfall events that trigger and no trigger shallow landslides, which is reflected in high performances (Andes 3, Amazon 1, Amazon 3 and Pacific 1 regions). However, in other regions, this separation between rainfall events is more complex to carry out, since there are more rainfall events with high magnitudes that do not trigger landslides but that exceed the thresholds, reflecting in lower performances (Andes 1, Andes 4 and Amazon 2). Thus, we could assume that in these regions there is a greater incidence of lithology and geology in the occurrence of SL than just the rains."

L. 329: high sensitivity to what?

Comment response: Thanks for the observation. High sensitivity to the little data, in the context of scare data of shallow landslide events of Peru. The text was edited for better understanding, as you can see:

"However, the uncertainties associated with these databases are the main source of uncertainty fo the thresholds. The few landslides recorded made that the validation performance had highly sensitive to the few data (i.e., a single event could lead to a high or low value of the performance statistics)."

Figure 7: Is there a reason for showing sensitivity/specificity? Wouldn't it be easier to interpret if you would just colour according to TSS?
This figure is very interesting and shows high spatial variability in the thresholds. Can you say something about this variability? E.g. is the threshold higher in wet regions? See e.g.
Leonarduzzi et al. (2017) Figure 7 or Marc et al. (2019).
Marc, O., Gosset, M., Saito, H., Uchida, T., Malet, J.P., 2019. Spatial Patterns of Storm- Induced Landslides and Their Relation to Rainfall Anomaly Maps. Geophys. Res. Lett.
167–177. https://doi.org/10.1029/2019GL083173

Comment response: Thanks for the observation. Our reason for showing the sensitivity/specificity was to show which parameter had a greater incidence in the TSS, whether it was the good detection of triggering events (sensitivity) or the good detection of non-triggering events (specificity).

We have taken into account your observations and recommendations and have included them in the discussions of the new version of the mn, as you can see below:

"Regarding the variability of the thresholds, we can explain it mainly to the rainfall climatology in Peru. It can be seen that the magnitudes have a relationship with respect to the spatial distribution of rainfall in Peru, that is, low thresholds related to rainfall of lesser magnitude in the arid zones in the western part of Peru (Pacific region), thresholds intermediates related to

the increase in the magnitude of rainfall in the middle part or mountainous region (Andes region) and the highest thresholds related to wet regions (Amazon region). However, the Andes 1, Andes 3 and Andes 6 regions do not have this relationship, so this discussion is not conclusive and is considered to be related to limited data, so it is suggested that this variability be discussed in future research that include more shallow landslides events data."

---

## Author Comment (AC2)

**Response to comment on nhess-2022-199**

The manuscript presents an interesting application of methods for the definition of empirical rainfall thresholds for landslide occurrence at a national scale. The aim of the paper is clear and the results are also well-presented. Despite some points not very clear, I found the manuscript clear and sufficiently well-organized. From a methodological point of view, I found some problems in the work, which should be addressed before the paper can be reconsidered for publication.

I list in the following some general comments and a few specific technical corrections and other suggestions.

Comment response: Thank you very much for your review, we have tried to make it more clear and correct the problems in the work considering your comments, corrections and suggestions in the new version of the manuscript. We are very grateful and sure that each of your comments contributed to the improvement of our work. Additionally, this document is highly important for the scientific community related to landslides in Peru since this type of work has not been developed in Peru, which, in addition, faces the limited availability of data compared to other countries. Lastly, other investigations also faced similar difficulties (e.g., Kirschbaum et al., 2015; Abraham et al., 2019). In this sense, this study is the first to be carried out on a national scale in Peru and its objective is to support the operational monitoring system of shallow landslides in Peru (https://www.senamhi.gob.pe/?p=monitoreo- silvia), and since our institution (SENAMHI) is responsible for maintaining this system, this work will contribute to giving it scientific validity, understanding its limitations but which will continue to be improved over time.
* * *
**Response to General comments**

The main problem of the work lies in the validation procedure. In particular, the use of only one year of data as validation set is inconvenient. This choice was proved to be not effective cause is too much linked to the variability of the selected year. Indeed, you found that the performances decreased in the validation, "which may be due to the fact that, in the year 2020, there were no extreme rainfall events as in other years, and the number of landslides was lower than in other years". A more reliable procedure would consider a random selection of triggering and non-triggering rainfall conditions in a calibration (e.g. 80% of the total) and a validation set (remaining 20%). You can found examples in: https://doi.org/10.1007/s11069-019-03830-x or https://doi.org/10.5194/hess-25-3267-2021

Comment response: Thank you very much for the observation, this is one of the discussions that we added taking into account your comments and observations. We take this methodology that has already been used in other investigations (e.g., Dikshit et al., 2019; Kirschbaum et al., 2015), however, as we conclude, it did not obtain good results for few data, so we add this discussion so that be taken into account in future research in Peru. We have added your observation to the new version of the manuscript, as you can see below:

"The calibration/validation methodology, based on take one year of observations for validation set, which was used in other research works (e.g., Dikshit et al., 2019;

Kirschbaum et al., 2015), is quite short and there is the risk of overinterpretation. It is therefore highly recommended for future research to expand the dataset and explore other calibration/validation methods, for example, a random selection of the differentiated data set for the calibration and validation (e.g., 70% for calibration and 30% for validation) (Brunetti et al., 2021; Gariano et al., 2020)."

In addition, in our future research we hope to advance in these limitations in Peru, for example, our perspective is to expand the database, for which we are working with INDECI (entity in charge of the attention of the population when landslides occur) for future studies that include greater data extension and include the random selection of the dataset.

--

The use of daily rainfall data is also not the best choice for defining rainfall thresholds, particularly for shallow landslides, given the high uncertainties related to this temporal resolution as highlighted by https://doi.org/10.1007/s1106 9-018-3508-4 https://doi.org/10.1007/s11069-019-03830-x. This should be pointed out and discussed better. I would add that there are currently other satellite-based rainfall products with better temporal resolutions (e.g GPM), which could be employed in such analyses.

Comment response: Thanks a lot for the observation. We agree that more exact thresholds could be defined with sub-daily rainfall data, however we chose to use these daily rainfall data for different reasons, the first is that this work is the first approximation of regional rainfall thresholds in Peru from from which new and better thresholds will be generated, in addition to the fact that we take into account different investigations that developed thresholds from daily rainfall data (e.g. (Berti et al. 2012; Kirschbaum and Stanley 2018; Leonarduzzi and Molnar 2020; Leonarduzzi, Molnar, and McArdell 2017; Monsieurs et al. 2019), in addition to the fact that these thresholds have the objective of improving landslide monitoring services triggered by rainfall that already exists in Peru (https://www.senamhi.gob.pe/?p=monitoreo-silvia) and that our institution, SENAMHI (the hydrometeorological service of Peru), is responsible for monitoring and improving it. Finally, until 2017 we used TRMM data for our hydroclimatic services, however, for a period of time the TRMM data was not maintained, and all our hydrological services that depended on this data had to stop, for this reason, at SENAMHI we choose to generate operational data (PISCO) that takes into account the assisted climatology data (e.g. PISCO monthy mean) but does not depend of external data base.

Currently, as SENAMHI  we are also focused on the generation of hourly rainfall product (e.g. Huerta et al., 2017 https://doi.org/10.1016/j.dib.2022.108570), but that it be updated in real time for our monitoring services, with which our next investigations regarding thresholds will take into account these hourly data.

--

The whole section 2.4 misses several information and needs a check and a huge review.

Comment response: Thanks a lot for the observation. We rewrite and reorganize the entire section for better understand, as you can see below.
"2.4 Rainfall threshold model

[revised manuscript text omitted]

It is not clear how the association between a rainfall event and a landslide is done (Line 120), in order to classify an event as a triggering rainfall event.

Comment response: Thanks for the observation. A rainfall event it considered as triggering event if during the duration of the rainfall event a shallow landslide was occurred. We edit the sentence for better understand, as you can see below.

"A rainfall event is considered a rainfall trigger event if it is associated with a landslide event, i.e., if during the duration of the rainfall event a shallow landslide was occurred."

Moreover, at line 118 it is reported that "For coastal Pacific regions, 0.5 mm was considered the minimum rainfall threshold". What about the other regions?

Comment response: Thanks for the observation. The minimum rainfall threshold considered for other regions is 1 mm, and only for the coastal Pacific region is 0.5 mm. We edit the equation and sentence for better understanding, as you can see below.

"… where s is the average of simple bias when rainfall stations reported a value of 0 rainfall compared with the estimation in PISCOpd_Op. And U0 is the initial minimum rainfall threshold, and it stablished as 1 mm for all region with exception of coastal Pacific regions which is considered 0.5 mm."

At lines 131-136, it is not clear the actual method used to define the thresholds, based both on 1 or 2 variables. How the parameters and the equations were obtained? Before "maximizing predictive performance" a threshold should be calculated using a method. Which method was used? This issue needs to be better explained.

Comment response: Thanks for the observation. The paragraph was edited and corrected for better explain, as you can see below:

"... Rainfall thresholds were established by maximizing TSS predictive performance in two ways: the first way includes every rainfall event property independently (Imax, E, D or Imean), and the second one determined was through curve-like thresholds that related two properties (Imax − D, E − D, Imean − D) in the form of $V = a.D^{-b}$, where V represents the rainfall properties (Imax, E, and Imean); a and b are the scale and shape parameters of the curve (while for logarithmic space, a is the intersection parameter and b denotes the slope of the linear curve). The approximation of the first form, thresholds based on only one of the rainfall event properties (Imax, E, D or Imean), was estimated whit the minimum radial distance to the perfect classificatory test (TSS=1, with se=1 and 1-sp=0) from the ROC space (e.g., Uwihirwe et al.; Gariano et al.) and the approximation of the second form, curve-like thresholds, was established with the optimization of a and b parameters of the curve model ($V = a.D^{-b}$) with an initial approximation of the curve based on a=average of the variable V of the triggering rainfall events and b=0. ...."

Moreover, at line 133 is written "variables independent of rainfall properties (Imax,E,D, Imean)"; actually, Imean and D are not independent on each other, being Imean=E/D. Please explain also this point.

Comment response: Thanks for the observation. We refer as variable independent to only one of the rainfall event properties (Imax, E, D, Imean). The sentence was corrected for better understand as you can see below:

"… the first way includes every rainfall event property independently (Imax, E, D or Imean), …"

Finally, I believe that proposing thresholds based only on one parameter (e.g. E, D, Imean, or Imax) is now anachronistic, given the huge literature on rainfall thresholds based on two variables.

Comment response: Thanks for the observation. We agree that there is a large amount of literature on thresholds based on two variables, although there is also literature that evaluates one-parameter variables and/or how they impact when combined with other variables (Hirschberg et al. 2021; Leonarduzzi et al. 2017 ; Uwihirwe, Hrachowitz, and Bogaard 2020), in this sense, our approach, being a novel work in Peru, is to provide variables that could be beneficial for certain regions and in future research combine or improve them in greater detail. Additionally, this paper is highly important for the scientific community related to landslides in Peru since this type of work has not been developed in Peru, which, in addition, faces the limited availability of data compared to other countries.

Regarding the thresholds based on two variables, actually there is no need to calculate both E-D and Imean-D thresholds, given that they are analytically equivalent, being Imean=E/D. I can't figure out how different results are obtained for the two types of thresholds (I-D and E-D); they should have the same performaces).

Comment response: Thanks for observation. We have taken into account your comment. According to the way we have defined the variables for a dataset, Imean, that is affected by D, does not have the same distribution as E. For example, two events with the same E (e.g. E=10), can have different D (e.g. D equal to 2 and 4 days), therefore, the Imean of both resulting events are different (Imean equal to 5 and 2.5 respectively), so the threshold could not be defined as the division of both. The next Fig. X1 shows what is mentioned for an example dataset, where it is observed that E and Imean have different density distributions and therefore their predictive potentials also change (i.e., the thresholds do not have the same Imean relationship =E/D).

[Figure]

Fig. X1: Density plot of the variables E (a), Imean (b) and D (c) for the same data set, where it is observed that the distributions of the variables E and Imean are different.

--

Line 170: actually, a threshold is represented by a point in the ROC space (the point is the TRP, FPR couple), so I believe that the area under curve is only a quadrangle. Please explain better this point. Being the thresholds represented only by one point in the ROC space, I would suggest using the distance of this point from the perfect classificatory point (upper left corner of the space, TPR=1, FPR=0) instead of the area under curve. You can find more details in https://doi.org/10.1016/j.geomorph.2014.10.019

Comment response: Thanks a lot for the suggest and clarification of the topics. We use this method to estimate the thresholds of one variable, and we edit and correct the paragraph and explanation of the calibration methods for better understanding, as you can see below:

"For thresholds based on rainfall event properties independently (Imax, E, D or Imean), the overall impression of the predictive power was estimated whit the so-called receiver operating characteristic (ROC) curve (Fawcett, 2006), from which the minimum radial distance to the perfect classificatory test (TSS=1, with se=1 and 1-sp=0) was used to select the individual variable threshold (e.g., Uwihirwe et al.; Gariano et al.) …"

Lines 179-182: actually, more simple and useful definitions are: TPR = TP/(TP + FN); FPR= FP/(FP + TN). I would suggest using these definitions instead of mentioning sensitivity and specificity.

Comment response: Thanks for the suggest. We use these definitions as we review that they were also used in many other current publications, but we have edited and added these citations for better understanding, as you can see:

"Some of the most common measures for landslide forecasting are the sensitivity (se = TP/(TP + FN )), specificity (sp = 1 − FP/(FP + TN )) and true skill statistic (TSS = se + sp −1) (e.g., Staley et al., 2013; Gariano et al., 2015; Leonarduzzi et al., 2017; Mirus et al., 2018; Leonarduzzi and Molnar, 2020; Hirschberg et al., 2021).

… The benefit of using the specificity over the false positive rate (FPR=FP/(FP+TN)) is that in a perfect model TSS, sensitivity and specificity all equal 1 (Hirschberg et al., 2021)."

Passing to Section 3, regarding the regionalization, it is not clear how many empirical points are employed for calculating the thresholds in each of the 11 regions. Please add this information and discuss possible limitations in case of thresholds based on too few points.

Comment response: Thanks for the observation. We add this information, and add a discuss on the new version of the manuscript, as you can see below:

"Hirschberg et al. (2021) found that 25 events are enough to limit the uncertainties in the ID threshold parameters to ±30% in his study, based on this, it is observed that there are several regions (Andes 3, 5, 6 and Amazon 1, Amazon 3 and Pacific 2) that do not exceed that quantity, so these regions have a greater source of uncertainty due to the quantity of the data. A summary of the number of shallow landslide events used for the research and the thresholds with best performances per region is presented in Table

3."

**Table 3.** Number of SL events and best thresholds for one and two variables for each region (Th: threshold, SL: number of landslides per region, Cal: Calibration, Val: Validation)

| Region | SL total | SL Cal | SL Val | Best Th - 1 variable | TSS | Best Th - 2 variables | TSS |
|---|---|---|---|---|---|---|---|
| Pacific 1 | 46 | 43 | 3 | $I_{max}$ | 0.68 | $I_{max} - D$ | 0.71 |
| Pacific 2 | 27 | 20 | 7 | $I_{mean}$ | 0.61 | $I_{mean} - D$ | 0.61 |
| Andes 1 | 34 | 28 | 6 | $I_{mean}$ | 0.43 | $I_{mean} - D$ | 0.44 |
| Andes 2 | 98 | 83 | 15 | $E$ and $I_{mean}$ | 0.58 | $I_{max} - D$ | 0.64 |
| Andes 3 | 17 | 10 | 7 | $I_{max}$ | 0.92 | $I_{max} - D$ | 0.91 |
| Andes 4 | 65 | 54 | 11 | $E$ | 0.51 | $I_{mean} - D$ | 0.52 |
| Andes 5 | 14 | 7 | 7 | $E$ | 0.67 | $I_{mean} - D$ and $E - D$ | 0.66 |
| Andes 6 | 4 | 3 | 1 | $D$ | 0.68 | $E - D$ | 0.65 |
| Amazon 1 | 6 | 6 | - | $I_{mean}$ | 0.74 | $I_{mean} - D$ | 0.77 |
| Amazon 2 | 54 | 41 | 13 | $E$ | 0.57 | $E - D$ | 0.58 |
| Amazon 3 | 12 | 10 | 2 | $E$ | 0.68 | $I_{mean} - D$ and $I_{max} - D$ | 0.73 |

Figure 6. Please note that the thresholds should have duration ranges based on the minimum and maximum durations of the triggering events. Theoretically, you can't draw a threshold in a duration value when you don't have a triggering event. This allow also avoiding having very low values of thresholds at long durations (see thresholds for Andes 4, 5, 6). Moreover, I would suggest correcting all the equations replacing Y and X with Imean and D, and replacing the "^" with a proper superscript.

Comment response: Thanks for the observation. The figure was edited taking in account your suggestions on the new version of the manuscript as you can see below:

[Figure]

Figure 7. Is there some physical explanation for the variation of the values of the 1-variable thresholds? In some cases, I see differences that seem not related to morphology or other environmental factors.

Comment response: Thanks for the observation. We have taken into account your observations and recommendations and have included them in the discussions of the new version of the manuscript, as you can see below:

"Regarding the variability of the thresholds, we can explain it mainly to the rainfall climatology in Peru. It can be seen that the magnitudes have a relationship with respect to the spatial distribution of rainfall in Peru, that is, low thresholds related to rainfall of lesser magnitude in the arid zones in the western part of Peru (Pacific region), thresholds intermediates related to the increase in the magnitude of rainfall in the middle part or mountainous region (Andes region) and the highest thresholds related to wet regions (Amazon region). However, the Andes 1, Andes 3 and Andes 6 regions do not have this relationship, so this discussion is not conclusive and is considered to be related to limited data, so it is suggested that this variability be discussed in future research that include more shallow landslides events data."
* * *
**Response to technical corrections and suggestions**

Abstract: I would use the present tense in the abstract

Comment response: Thanks for the suggest. It was edited on the new version of the manuscript, as you can see:

"Abstract. The objective of this work is to generate and evaluate regional rainfall thresholds obtain from a combination of high-resolution gridded precipitation data (PISCOpd_Op), developed by the National Service of Meteorology and Hydrology of Peru (SENAMHI), and information from observed shallow landslide events. The landslide data were associated with rainfall data, determining triggering and non-triggering rainfall events with rainfall properties from which rainfall thresholds are determined. The validation of the performance of the thresholds is carried out with events that occurred during 2020 and focus on evaluating the operability of these thresholds in landslide warning systems in Peru. The thresholds are determined for 11 rainfall regions. The method of determining the thresholds is based on an empirical–statistical approach, and the predictive performance of the thresholds is evaluated whit the "true skill statistics" (TSS). The best predictive performance is the mean daily intensity-duration ($I_{mean} - D$) threshold curve, follow by accumulated rainfall E. This work is the first attempt to estimate regional thresholds on a country scale in order to better understand landslides in Peru, and the results obtained reveal the potential of using thresholds in the monitoring and forecasting of shallow landslides caused by intense rainfall and in supporting the actions of disaster risk management."

I would use rainfall instead of precipitation everywhere in the text.

Comment response: Thanks for the suggest. We use rainfall instead of precipitation on the new version of the manuscript.

Line 24: "Terrain saturation is the original cause of landslide occurrence". Actually, this depends on the type of landslides.

Comment response: Thanks for the observation. This sentence it was removed for better understand according the comment.

Line 33: perhaps the correct reference is Segoni et al 2018 (already mentioned), not Segoni et al 2014

Comment response: Thanks for the observation. The correct reference is Segoni et al., 2018, and it was corrected in the new version of the manuscript.

Line 36-39: The sentence "For example, global thresholds have been developed based on antecedent precipitation indices (Caine, 1980; Guzzetti et al., 2008; Kirschbaum and Stanley, 2018), and national thresholds have been established under an empirical–statistical approach (Leonarduzzi et al., 2017; Peruccacci et al., 2017a; Uwihirwe et al., 2020)." is not correct. Actually, both the mentioned thresholds based on antecedent precipitation and the cited national thresholds are established using and empirical-statistical approach. Please review and correct.

Comment response: Thanks for the observation. The text has been rephrased in order to clarify the main idea, as you can see below.

"For example, there is been developed empirical–statistical approach to the estimation of global thresholds (Caine, 1980; Guzzetti et al., 2008; Kirschbaum and Stanley, 2018), and national thresholds (Leonarduzzi et al., 2017; Peruccacci et al., 2017a; Uwihirwe et al., 2020)."

Line 38: Note that there are two references to the work "Peruccacci et al. (2017)" a and b, which are actually the same.

Comment response: Thanks for the observation. The reference has been corrected.

Line 47: I would suggest using "slope" instead of "hillside"

Comment response: Thanks for the observation. The text has been changed, as you can see below.

"Thresholds can be set for different spatial scales depending on the extent of the analysis, and these can be categorized into six classes: global, national, regional, basin, local, and slope scales. …"

Line 51: in relation to environmental subdivisions within a national territory, please consider also the work of Peruccacci et al. (2017) – already mentioned – which present several morphological, geological, meteorological, climatic subdivision of the Italian territory with the aim of defining rainfall thresholds.

Comment response: Thanks for the suggestion. The reference was added, as you can see below.

"…, as well as environmental subdivision within a national territory based on erodibility and climatology represented by the maximum daily intensity of a rainfall event (Leonarduzzi et al., 2017) or on topography, lithology, land-use, land cover, climate, and meteorology (Peruccacci et al., 2017)."

Caption of figure 2. Delete "Methodology six steps"

Comment response: Thanks for the observation. The caption has been corrected.

Line 101: I suppose you wanted to write "is shown in Figure 2".

Comment response: Thanks for the observation. It is Map Figure (Fig. 2), and it was edited on the new manuscript.

Line 185-186: please check syntax and grammar.

Comment response: Thanks a lot for the observation. I checked the syntax and grammar and simplified the paragraph, as you can see below.

"For thresholds based on independent variables (Imax, E, D, Imean), the overall impression of the predictive power was estimated whit the so-called receiver operating characteristic (ROC) curve (Fawcett, 2006), from which the minimum radial distance to the perfect classificatory test (TSS=1, with se=1 and 1-sp=0) was used to select the individual variable threshold (e.g., Uwihirwe et al.; Gariano et al.) while for the threshold curve (Imax−D, E−D, Imean−D) the scale parameter a and the shape parameter b are simultaneously tuned to maximize the the true skill statistics (TSS) (e.g., Leonarduzzi et al.; Hirschberg et al.). This maximization was carried out automatically using the shuffled complex evolutionary algorithm (SCEA-UA) (Duan et al., 1993), considering the TSS as the objective function. The methodology was applied for each region within the analysis area, finding different thresholds for each of them."

Line 196: actually, TSS varies between -1 and 1, as you correctly mentioned some lines above.

Comment response: Thanks a lot for the observation. It was edited in the manuscript.

Table 3: I would suggest using always the same number of decimal places.

Comment response: Thanks a lot for the suggestion. It was edited in the manuscript using 2 decimal places.

---

## Author Comment (AC3)

**Response to comment on nhess-2022-199**

**Response to general comments**

The manuscript deals with the development and evaluation of regional landslide precipitation thresholds in Peru. The Authors used the available high-resolution gridded precipitation and landslide events data to define empirical thresholds which is an important step towards the development of landslide early warning system in Peru (a country with limited landslide studies).

The study seems very important especially in a country with limited landslide studies yet with frequent landslide hazards problems. However, some sections of the manuscripts need to be polished for a better flow of the manuscript. Some points also need to be corrected:

Comment response: Thank you very much for your review and general comments, we have tried to make it not a bit difficult to read and also not seems unorganized, considering all your comments in the new version of the manuscript. Additionally, this document is highly important for the scientific community related to landslides in Peru since this type of work has not been developed in Peru, which, in addition, faces the limited availability of data compared to other countries. Lastly, other investigations also faced similar difficulties (e.g., Kirschbaum et al., 2015; Abraham et al., 2019).

**Specific comments**

**Section** 2 This section presents the methodology used. Figure 1 summarises the methodology in 6 steps which is really good. However, from sub_sect. 2.1 to 2.6 one would expect the details from step 1 to step 6. These steps are not outlined clearly in these sections and may break the flow of the manuscript not only in Methodology section but also the Results section.
Comment response: Thanks for the observation. This observation was taken in account in the new manuscript, we reorder and organized the methodology on the sub sec 2.4 as you can see below:

"2.4 Rainfall threshold model

[revised manuscript text omitted]

**Minor comments/technical corrections**

Figure 2 caption. "Methodology six steps" is not relevant for the Figure. I would suggest to correct the Caption as: "Study area. Left: Spatial distribution of the Global Landslide Catalog (red) and SENAMHI landslide inventory (yellow). Right: Eleven landslide-susceptibility regions for Peru and distribution of calibration (blue) and validation (yellow) landslides" .

Comment response: Thanks for the suggest. It was added on the new version of the manuscript, as you can see:

"Study area. Left: Spatial distribution of the Global Landslide Catalog (red) and SENAMHI landslide inventory (yellow). Right: Eleven landslide- susceptibility regions for Peru and distribution of calibration (blue) and validation (yellow) landslides."

LL101. …. Is shown in 3. There is something missing. Is it Figure 2? Or sect. 3?
Comment response: Thanks for the observation. It is Map Figure (Fig. 2), and it was edited on the new manuscript.

LL126-127. "If it is possible to forecast or warn of possible landslides". To be corrected as "If it is possible to forecast or warn landslides"

Comment response: Thanks for the observation. It was corrected on the new version of the manuscript, as you can see:

"The reason for analyzing the second scenario was to evaluate the level of incidence that is attributed only to antecedent conditions for landslide occurrence, as this allows us to evaluate if it is possible to forecast or warn landslides based only on the antecedent conditions."

LL 131. " triggering rain evens"   to be corrected as "triggering rain events"

Comment response: Thanks for the observation. It was corrected on the new version of the manuscript, as you can see:

"… objectively select a rainfall threshold that separates triggering rainfall events from non-triggering rainfall events with the best level of predictive performance."

Figure 7 caption is a little bit messy. May be this: The first column shows the spatial distribution of Rainfall thresholds for independent variables magnitude for Peru: (a) D (days), (b) total cumulative rainfall E (mm), (c) mean daily intensity $I_{mean}$ (mm/day) and (d) maximum daily intensity Imax (mm/day). The second and third columns show the bivariate maps indicating the spatial distribution of the sensibility (probability of correctly predicting landslide triggering rainfall events) and specificity (probability of correctly predicting non-triggering rain events from landslide) of the thresholds for calibration and Validation.

Comment response: Thanks a lot for the observation and recommendation. It was corrected on the new version of the manuscript, as you can see:

"Figure 7. The first column shows the spatial distribution of Rainfall thresholds for independent variables magnitude for Peru: (a) day D (days), (b) total cumulative rainfall E (mm), (c) mean daily intensity Imean (mm/day) and (d) maximum daily intensity Imax (mm/day). The second and third columns show the bivariate maps indicating the spatial distribution of the sensitivity (probability of correctly predicting landslide triggering rainfall events) and specificity (probability of correctly predicting non-triggering rainfall events from landslide) of the thresholds for calibration and validation."

---

## Author Response (AR2)

**Response to Report #1 - Anonymous Referee #1**

GENERAL COMMENTS

The quality and readability of the manuscript has significantly improved compared to the first version. It's very clear that the authors made an effort in revising their work based on the reviewer's comments. There is a better structure and most ambiguities have been sorted out. I still have one relevant general comment and some specific comments below which I think should be considered before publication in NHESS. Additionally, I suggest that the manuscript is edited by a native English speaker or maybe just by using Grammarly, because the grammar and vocabulary have room for improvement. I made some corrections but it's probably not comprehensive.

**Comment response:** Thank you very much for your review, we learned a lot from your comments and the manuscript was significative improved. We edit and use Grammarly in the new version of the mn.

Regarding the ID vs ED thresholds, I am not convinced by your argument on the differences in distribution of variables E and I. Sure, you're right, but this only explains why I and E have different performances on their own, not why ED and ID should be different. When you do the math, you can rewrite I=aD^b to E=aD^(b+1). When you transform from ID to ED plot, and look at rainfall events with the same duration (parallel to y-axis), these points will just scale with duration, but the information or order of the data does not change by scaling with a dependent variable (D). This is maybe not the perfect explanation (maybe another reviewer can do better) but I made a quick test with both ED and ID and the result of the threshold is exactly of the relation above and the performance stayed the same. If you're still convinced you're right, maybe do an example of ID and ED thresholds with a strongly reduced dataset to show that the result is different (or something similar). If I am right, the results should be checked carefully.

**Comment response:** Thanks for the observation. We agree with you and your explanation was perfect for us. We did a rapid example (Fig. X1) and we understand that I=aD^b are equivalent to E=aD^(b+1), but this occurred when the shape parameter is obtained by linear regression and the scale parameter is selected by other methods (e.g., frequentist method). In our study, as we mentioned, we optimized both parameters a and b ("*this optimization was automatically calibrated using the shuffled complex evolutionary algorithm (SCEA-UA) (Duan et al., 1993), considering the TSS as the objective function.*"), so it´s not necessary to coincide with the regression, we assume that there is a threshold that not adjust necessary with the linear regression.

[Figure]

Fig. X1. Example of I=aD^b and E=aD^(b+1) where the parameter b its yellow highlight.

SPECIFIC COMMENTS

L1: I would avoid using abbreviations like PISCOpd_Op in the abstract
**Comment response:** Thanks for the observation. We avoid using abbreviations in the new version of the mn.

L1: obtained
**Comment response:** Thanks for the observation. It was edited in the new version of the mn.

L9: followed
**Comment response:** Thanks for the observation. It was edited in the new version of the mn.

L25: "flow in channels, streams and rivers" I would just say "stream flow"
**Comment response:** Thanks for the observation. It was edited in the new version of the mn.

L36: change to "…, empirical-statistical approaches for the estimation of global (citations) and national (citations) thresholds been developed."
**Comment response:** Thanks for the observation. It was edited in the new version of the mn.

L38: to forecast or for forecasting
**Comment response:** Thanks for the observation. It was edited in the new version of the mn.

L42: "Empirical approaches are widely applied …"
**Comment response:** Thanks for the observation. It was edited in the new version of the mn.

L48: This is kind of repetitive because you say that in L35 already
**Comment response:** Thanks for the observation. It was deleted in the new version of the mn.

L58-59: is this decision based on the findings of other studies?
**Comment response:** Thanks for the observation. Yes, based on part of the methodology of Leonarduzzi et al., 2017, and a previous work developed on SENAMHI (Yupanqui et al., 2017).

L60: is it really for the purpose of monitoring? Not to test the feasibility of a potential early-warning system?
**Comment response:** Thanks for the recommendation. It was edited in the new version of the mn, as you can see below.

*"The main objective of this work is to estimate rainfall thresholds to test the feasibility of a potential early-warning system of shallow landslides generated by rainfall from a gridded rainfall database and shallow landslide inventory."*

L61-62: I think it's rather "…implementing an objective methodology for empirical rainfall-based landslide early warning at a national scale" or something like that
**Comment response:** Thanks for the recommendation. It was edited in the new version of the mn, as you can see below.

*"Additionally, this work focuses on implementing an objective methodology for empirical rainfall-based landslide early warning at a regional scale combining a gridded rainfall database and shallow landslide inventory."*

L75-80: I don't understand what the susceptibility map and the basin map were used for. Before you said you said that regions are discretized by max rainfall (L59).
**Comment response:** Thanks for the observation. These lines are just the explanation of the delimitation of the study area (used by SENAMHI for the landslide monitoring service). This area was discretized by max rainfall.

L85: is reference ET the same as potential ET?

**Comment response:** Thanks for the observation. The **reference evapotranspiration** (ETo) is the evapotranspiration rate of a reference surface (a hypothetical grass reference crop with specific characteristics) which occurs without water restrictions (Allen, R. G. et al. 1998). For more information about the differences between reference evapotranspiration (ETo) and potential evapotranspiration (ETp) you can see Xiang et al. 2020 (Similarity and difference of potential evapotranspiration and reference crop evapotranspiration – a review - https://doi.org/10.1016/j.agwat.2020.106043).

L86: how much is 0.1° approx. in m or km?
**Comment response:** Thanks for the observation. 0.1° is approx. ~10 km.

L100: there's a repetition here with regard to the certainty in recent years
**Comment response:** Thanks a lot for the observation. It was deleted in the new version of the mn.

L103: what do you mean by "geospatial analysis" and based on what was the one event excluded?
**Comment response:** Thanks a lot for the observation. It refers to the use of geospatial tools: spatial sub-setting. It was changed for better understanding.

Figure2: in sttep1 "don't trigger" instead of "no trigger"

[Figure]

Figure3b: Please draw a clear line for Imean and E. Is Imean the green or the red dashed line? It's not clear to me what the green line is around the bars. E should be much higher as it's the sum of the six bars, shouldn't it?
**Comment response:** Thanks a lot for the observation. Yes, Imean is the green line that involves all the event representing the sum of the six bars, and for better understanding, we add a new blue line for Imean. The new version of the figure is shown below.

[Figure]

L129: "…, they were classified into triggering and non-trigger events, i.e. if a landslide occurred during the rainfall event."
**Comment response:** Thanks for the observation. It was edited in the new version of the mn.

L134-137: Maybe I'm misunderstanding something. Maybe you could draw these two scenarios in Figure 3 so it becomes clear. One scenario is just the daily rainfall (independent from the events you defined) and the other is the defined event minus the day of landslide occurrence, right? Why is not one scenario the defined event including the day of landslide triggering?
**Comment response:** Thanks for the observation. It was edited in the new version of the mn for better understanding.

"The first scenario (entire event - EE) considers all the rainy days of the rainfall event including the rainfall of the landslide occurrence day to determine the properties of the rainfall event (Figure 3). The second scenario (antecedent event - AE) considers only the antecedent rainy days of landslide occurrence to determine the properties of the rainfall event, i.e., AE does not consider the rainfall of the landslide occurrence day."

L141: dividing
**Comment response:** Thanks for the observation. It was edited in the new version of the mn.

L150: Why do you use different scores for uni- and multivariate predictors?
**Comment response:** Thanks for the observation. It's because we use different methodologies to define the thresholds, in the case of univariate predictors, the thresholds it was selected directly from the ROC space, and for multivariate predictors, we use an automatic optimization, which needs an optimization function based on TSS.

L154: applying
**Comment response:** Thanks for the observation. It was edited in the new version of the mn.

L188: change to "…a confusion matrix was used…"
**Comment response:** Thanks for the observation. It was edited in the new version of the mn.

L210: the year is missing in the citation
**Comment response:** Thanks for the observation. It was edited in the new version of the mn.

L287-291: This last paragraph I think should go into the discussion section. The results overall read very well!
**Comment response:** Thanks a lot for the recommendation. It was edited in the new version of

the mn.

L294: Technically, because your thresholds are based on TSS and radial distance, the thresholds are not only based on rainfall events associated with landslides, but also with non-triggering rainfall events
**Comment response:** Thanks a lot for the recommendation. It was edited in the new version of the mn.

L299: 2x shown
**Comment response:** Thanks a lot for the recommendation. It was edited in the new version of the mn.
"The estimated thresholds are shown in Table 1 for independent variables and Table 2 for curve thresholds."

L303: is the background rain the same as antecedent rain? I think you used this term earlier. Please be consistent.
**Comment response:** Thanks a lot for the recommendation. It was edited in the new version of the mn.
"However, it allows us to associate landslide events with the antecedent rain conditions of the last 8 days, an association that can be used for future research."

L322: …in their study. Based on…
**Comment response:** Thanks a lot for the recommendation. It was edited in the new version of the mn.

L334-339: I think it's good to have this listing but some points need to be more specific to avoid misunderstandings. Not necessarily more explanation, but more precise. ii) isn't this the same as i) because a 10 km cell may cover several streams, or what exactly do you mean? Iii) you mean because your landslide record may not be complete? Iv) because thresholds based on short records may still be uncertain? V) is unclear to me what is meant.
**Comment response:** Thanks a lot for the recommendation. It was edited in the new version of the mn.

"There are still many limitations to rainfall threshold study at the regional scale in Peru. Mainly the landslide short records are not enough to limit uncertainty in the threshold definition (Peruccacci et al., 2012; Hirschberg et al., 2021). Another important source of uncertainty was the use of coarse temporal rainfall data resolution that caused a systematic underestimation of the thresholds (Gariano et al., 2020; Marra, 2019). Another is the spatial rainfall data resolution because a 10 km cell may cover several streams. And finally, the regionalization can be not enough representative of the high variability of descriptor landslide variables. These limitations must be taken into account in future research."

L344: "relationship" instead of "interrelation"
**Comment response:** Thanks a lot for the recommendation. It was edited in the new version of the mn.

L346-349: very long sentence, consider splitting it in two.
**Comment response:** Thanks a lot for the recommendation. It was edited in all parts of the new mn.
"Daily gridded rainfall data and landslide data were used to estimate landslide-triggering and non-triggering rainfall events. With this data was possible to estimate and validate rainfall thresholds for the activation of shallow landslides triggered by rainfall."

L353: I would change "accumulated daily intensity" to "daily rainfall"
**Comment response:** Thanks a lot for the recommendation. It was edited in the new version of the mn.

L354: maybe "variability" instead of "differentiation"? and then "This differences in performance are…"
**Comment response:** Thanks a lot for the recommendation. It was edited in the new version of the mn.

L354-361: These lines are hard to read and understand. Shouldn't you regionalization improve the situation? Or is the regionalization not that good so within each region there is still high variability? What could you do about it in the future? What do you mean by greater incidence of lithology and geology, do these affect the thresholds or is it a question of sediment supply? What is SL (I would avoid abbreviation in conclusions)?
**Comment response:** Thanks a lot for the observation. Regionalization improves the climate context of Peru, but it is not conclusive, we cannot divide into more regions for the landslides little data. In some regions with lower performances, we think that other geological components influence the occurrence of shallow landslides, and future studies can explore new regionalization based on lithology (e.g., Peruccacci et al. 2011).

The conclusion was edited for better understanding in the new version of the mn, as you can see below.

*"The performances of the calibrated thresholds had a high variability between regions. These differences in performance are associated with the high variability of rainfall events in each region, where best performances occur in areas where it is easier to separate rainfall events that trigger and non-trigger shallow landslides (e.g., Andes 3, Amazon 1, Amazon 3 and Pacific 1 regions). However, in other regions, this separation between rainfall events is more complex to carry out, since there are a lot of rainfall events with high magnitudes that do not trigger landslides, reflecting in lower performances (e.g., Andes 1, Andes 4 and Amazon 2). Thus, the regionalization shows that exists regions where the climate component had more predominance in the shallow landslide occurrence in comparison with other regions where lithology could have more influence in the occurrence of shallow landslides than just the rains. Future studies can explore regionalization based on lithology."*

L362: you could add that despite these uncertainties, the framework you've set up allows for systematically updating the thresholds as the records grow.
**Comment response:** Thanks a lot for the recommendation. It was edited in the new version of the mn.
*"Despite these uncertainties, the framework set up of this work allows to systematically update the thresholds as the records grow."*

L372: Thumb up for making the code available!
**Comment response:** Thanks a lot for the comment.

**Response to Report #2 - Anonymous Referee #2**

Dear Authors,

my sincere apologies for my very late reply!

I've read your replies to my comments and the revised version of the manuscript. I found a relevant improvement from the original version, despite some limitations remains, as acknowledged by you.

**Comment response:** Thank you very much for your review, we learned a lot from your comments and the manuscript was significative improved. We hope that the answers in this document satisfy your observations.

The most crucial point is the number of empirical points used to calculate the thresholds in each pre-defined region. This number is depicted only now after my comment. Overall, looking at table 3, the number of points is low (except for Andes 2 and Andes 4, and to a minor extent Pacific 1 and Amazon 2). In some cases, the number of empirical points is not acceptable. The mentioned work (Hirschberg et al. 2021) refers to a little basin affected by debris flows, covering a few square kilometers. Therefore, I think it can't be taken as a reference, being the regions defined in this work very wide. Other works, on a regional scale (e.g. https://doi.org/10.1016/j.geomorph.2011.10.005) suggested higher values for obtaining reliable thresholds with acceptable uncertainties. Moreover, an uncertainty of 30% is too high in my opinion for an operational tool like the one proposed in the work. I suggest merging the regions into only three macro-regions (i.e. Pacific, Andes, and Amazon) in order to re-calculate the thresholds and obtain more reliable results. Alternatively, another possible merging might be: Pacific (former Pacific 1 and Pacific 2; 73 landslides); Andes 01 (former Andes 1 and Andes 2; 132 landslides); Andes 03 (former Andes 3, Andes 4, Andes 5, and Andes 6; 100 landslides); Amazon (former Amazon 1, Amazon 2, and Amazon 3; 72 landslides).

Other issues were solved, even if I think they deserve additional comment/discussion.

**Comment response:** Thanks for the observation. We applied the suggestion, merging Pacific (former Pacific 1 and Pacific 2; 73 landslides); Andes 01 (former Andes 1 and Andes 2; 132 landslides); Andes 02 (former Andes 3, Andes 4, Andes 5, and Andes 6; 100 landslides); Amazon (former Amazon 1, Amazon 2, and Amazon 3; 72 landslides).

Regarding, we noted that metrics worsen, so we opted, based on your comment, to taking account the 4 regions with a high number of events (Andes 2 and Andes 4, Pacific 1 and Amazon 2), and we are a pleasure to see that the metrics slightly better for validation procedure (Table X1).

**Table X1:** Comparison of TSS mean between different regionalization (11 reg: all regions, 4 macro-reg: suggested merge, 4 reg: 4 regions with a high number of events)

| Number of regions | Imean-D (TSS) | | Imax-D (TSS) | |
|---|---|---|---|---|
| | Cal | Val | Cal | Val |
| 11 reg | 65% | 42% | 62% | 42% |
| 4 macro-reg | 56% | 36% | 52% | 42% |
| 4 reg | 60% | 44% | 60% | 45% |

According to this comparison, we decided to add this to the discussion and the conclusions, as you can see below.

Discussion:

"… Peruccacci et al. (2012) found that the number of events must be mayor to 175 to limit the relative uncertainty below 10% but this figure may change for a different data set. Based on this, it is observed that only four regions (Andes 2, Andes 4, Pacific 1 and Amazon 2) have a number of events that are acceptable. The other regions have a greater source of uncertainty due to the quantity of the data. …"

Conclusion:

"Through the rainfall and landslides databases, it is possible to generate daily rainfall thresholds for shallow landslide occurrence. However, the uncertainties associated with these databases are the main source of uncertainty for the thresholds. The few landslides recorded made the validation performance highly sensitive to the few data (i.e., a single event could lead to a high or low value of the performance statistics). Thus, only four regions (Andes 2, Andes 4, Pacific 1 and Amazon 2) have enough events to limit these uncertainties. Despite these uncertainties, the framework set up of this work allows for systematic updates of the thresholds as the records grow."

Regarding the validation procedure, I think the provided reply is acceptable. However, I would suggest including the results of Table X2 in the discussion, also because they are relevant for the case study.

**Comment response:** Thanks for the suggestion. We add a brief discussion and Table X2 in the new version of the mn, as you can see below.

"The calibration/validation methodology, based on taking one year of observations for the validation set, which was used in other research works (e.g., Kirschbaum et al., 2015b; Dikshit et al., 2019), is quite short and there is the risk of overinterpretation. For this reason, this method was compared with other validation method based on a random selection of the data set (e.g., Brunetti et al., 2021; Gariano et al., 2020). According to this method, the data was divided into 70% for calibration and 30% for validation. The comparison of both validation approaches is shown in the Table 4. In this regard, the comparison between the validation methods did not indicate significant changes between each method. The results are very similar probably because the data size is not large enough to note the variations between the methods. It is highly recommended for future research focus in the expansion of the data-set and then compare the validation methods efficiency."

**Table 4**: TSS comparison summary between validation approaches

| Procedure | TSS comparison summary between two validation approaches: 1-year selection vs. random selection | | | | | | | | |
|---|---|---|---|---|---|---|---|---|---|
| | Imean-D | | | Imax-D | | | E-D | | |
| | 1 year | Random | ΔTSS | 1 year | Random | ΔTSS | 1 year | Random | ΔTSS |
| Calibration | 0.65 | 0.61 | -0.04 | 0.62 | 0.59 | -0.03 | 0.59 | 0.58 | -0.01 |
| Validation | 0.42 | 0.50 | 0.08 | 0.42 | 0.45 | 0.03 | 0.43 | 0.40 | -0.02 |

Regarding the use of rainfall data with the daily temporal resolution, I still think that even

a brief acknowledgment of this limitation is needed.

**Comment response:** Thanks for the observation. We add this brief acknowledgment of this limitation in the new version of the manuscript, as you can see below.

"There are still many limitations to rainfall threshold study at the regional scale in Peru. Mainly the landslide short records are not enough to limit uncertainty in the threshold estimation (Peruccacci et al., 2012; Hirschberg et al., 2021). Another important source of uncertainty was the use of coarse temporal rainfall data resolution that cause a systematic underestimation of the thresholds (Gariano et al., 2020; Marra, 2019). Additionally, the spatial rainfall data resolution of ~10 km cell may cover several streams. And finally, the regionalization can be not enough representative of the high variability of descriptor landslide variables. It is highly necessary that these limitations must be taken into account in future research."

Regarding the two-variables thresholds, i.e. E-D and Imean-D, I still think that there is no need for calculating both of them. I thank you for the analysis provided; however, it is related to a single-variable case. The literature on two—variables rainfall thresholds is full of examples showing that E-D and Imean-E are analytically equivalent. As an example, if the equation for an E-D threshold is E=a*D^(b), the equation for the Imean-D threshold is Imean=a*D^(1-b).

**Comment response:** Thanks for the observation. According to the observation of the two referees, we decided to exclude E-D in the new version of the mn.

Overall, a grammar and syntax check is needed, given that there are still some problems (e.g. lines 125-129; line 145 [rainfall trigger event]; line 250 [Rainfall–landslide threshold]; and so on…)

**Comment response:** Thanks for the observation. We changed [rainfall trigger event] by [triggering rainfall event], and [rainfall–landslide threshold] by [rainfall thresholds for landslides occurrence] in the new version of the manuscript. In addition, we check all the paper to improve the syntax.

Regarding the figures: In figures 2 and 3, precipitation still needs to be corrected into rainfall. In figure 3 please check the units of measurement. Figure 6 in the text is still the old version, not the revised one.

**Comment response:** Thanks for the observation. We edit the figures it in the new version of the manuscript.